# Efficacy and safety of passive immunotherapies targeting amyloid beta in Alzheimer's disease: A systematic review and meta-analysis

Reina Tonegawa-Kuji[1,2], Yuan Hou[1,2], Bo Hu[3,4], Noah Lorincz-Comi[1,2], Andrew A. Pieper[5,6,7,8,9], Babak Tousi[10], James B. Leverenz[10], Feixiong Cheng[1,2,4,11]*

1 Cleveland Clinic Genome Center, Lerner Research Institute, Cleveland Clinic, Cleveland, Ohio, United States of America, 2 Genomic Medicine Institute, Lerner Research Institute, Cleveland Clinic, Cleveland, Ohio, United States of America, 3 Department of Quantitative Health Sciences, Lerner Research Institute, Cleveland Clinic, Cleveland, Ohio, United States of America, 4 Department of Molecular Medicine, Cleveland Clinic Lerner College of Medicine, Case Western Reserve University, Cleveland, Ohio, United States of America, 5 Department of Psychiatry, Case Western Reserve University, Cleveland, Ohio, United States of America, 6 Brain Health Medicines Center, Harrington Discovery Institute, University Hospitals Cleveland Medical Center, Cleveland, Ohio, United States of America, 7 Geriatric Psychiatry, GRECC, Louis Stokes Cleveland VA Medical Center, Cleveland, Ohio, United States of America, 8 Institute for Transformative Molecular Medicine, School of Medicine, Case Western Reserve University, Cleveland, Ohio, United States of America, 9 Department of Neurosciences, Case Western Reserve University, School of Medicine, Cleveland, Ohio, United States of America, 10 Lou Ruvo Center for Brain Health, Neurological Institute, Cleveland Clinic, Cleveland, Ohio, United States of America, 11 Case Comprehensive Cancer Center, Case Western Reserve University School of Medicine, Cleveland, Ohio, United States of America

* chengf@ccf.org

## Abstract

### Background

While recently U.S. FDA-approved anti-amyloid beta (anti-Aβ) monoclonal antibodies (mAbs) offer new treatment approaches for patients suffering from Alzheimer's disease (AD), these medications also carry potential safety concerns and uncertainty about their efficacy for improving cognitive function. This study presents an updated meta-analysis of cognitive outcomes and side effects of anti-Aβ mAbs from phase III randomized controlled trials (RCTs) in patients with sporadic AD.

### Methods and Findings

Phase III randomized, placebo-controlled blinded trials evaluating the efficacy and safety of anti-Aβ mAbs in patients with AD were identified through a search in clinicaltrials.gov, PubMed and Embase on January 14th, 2024. The retrieved studies were further screened from January 15th, 2024, to February 14th, 2024. We included studies that had been published in any language. Quality of trials was assessed using the Jadad score and publication bias was assessed using Egger's test and Funnel plot. Primary outcomes were mean changes from baseline to post-treatment in Clinical Dementia Rating scale-Sum of Boxes (CDR-SB) and AD Assessment Scale-Cognitive Subscale (ADAS-Cog) scores, and

provided the original author and source are credited.

**Data availability statement:** Search results, data collected from each study and the R script used for the analysis are available on Github (https://github.com/ChengF-Lab/Meta_ADAb).

**Funding:** This work was primarily supported by the National Institute on Aging (NIA) under Award Number R01AG084250, R56AG074001, U01AG073323, R01AG066707, R01AG076448, R01AG082118, RF1AG082211, and R21AG083003, and the National Institute of Neurological Disorders and Stroke (NINDS) under Award Number RF1NS133812 to F.C. This work was supported in part by the National Institute on Aging under Award Number P30AG072959 to J.B.L. This work was supported in part by the Case Western Reserve University Rebecca E. Barchas MD, DLFAPA, University Professorship in Translational Psychiatry, the Valour Foundation, AHA/ Allen Initiative in Brain Health and Cognitive Impairment (Project 19PABH134580006), the Elizabeth Ring Mather & William Gwinn Mather Fund, S. Livingston Samuel Mather Trust, and the Louis Stokes VA Medical Center resources and facilities to A.A.P. This work was partly supported by the Alzheimer's Association award (ALZDISCOVERY-1051936) to F.C. The funders had no role in study design, data collection and analysis, decision to publish, or preparation of the manuscript.

**Competing interests:** I have read the journal's policy and the authors of this manuscript have the following competing interests: J.B.L has received consulting fees from consulting fees from Vaxxinity, grant support from GE Healthcare, in kind support from Amprion, and serves on a Data Safety Monitoring Board for Eisai. B.T, has received consulting fee from Eisai, Biogen, Novo Nordisk and Lilly. The other authors have no competing interests.

**Abbreviations :** AD, Alzheimer's disease; ADAS-Cog, AD Assessment Scale-Cognitive Subscale; anti-Aβ, Anti-amyloid beta; ARIA, Amyloid-related imaging abnormalities; ARIA-E, Amyloid-related imaging abnormalities with edema; ARIA-H, Amyloid-related imaging abnormalities with hemorrhage; CDR-SB, Clinical Dementia Rating scale-Sum of Boxes; CI, Confidence Interval; CSF, Cerebral Spinal Fluid; FDA, Food and Drug Administration; mAbs, monoclonal Antibodies; MCI, Mild Cognitive Impairment; MD, Mean Difference; MMSE, Mini-Mental State Examination; NNH, Number Needed to Harm; NNT, Number Needed to Treat; RCT, Randomized Controlled Trials; RR, Risk Ratio; SMD, Standardized Mean Difference.

secondary outcomes were adverse events, including amyloid-related imaging abnormalities with edema (ARIA-E), and ARIA with hemorrhage (ARIA-H). Random-effects meta-analysis and meta-regression analyses were conducted. The literature search identified 13 phase III RCTs, which included 18,826 patients with mild cognitive impairment or dementia due to AD. Compared with placebo, treatment with mAbs significantly improved cognitive performance on CDR-SB (mean difference –0.25, 95% confidence interval [CI] [–0.38, –0.11]) and ADAS-Cog (standardized mean difference –0.09, 95% CI [–0.12, –0.06]), in which a negative change indicates improvement for both scores. Meta-regression analysis suggested that trials enrolling patients with early-stage AD were associated with better efficacy. Elevated risk of ARIA-E (risk ratio [RR] 9.79, 95% CI [5.32,18.01]), ARIA-H (RR 1.94, 95% CI [1.47,2.57]), and headaches (RR 1.21, 95% CI [1.10,1.32]) were noted. Statistical heterogeneity was relatively high for ARIA-E and ARIA-H, leading to wide confidence intervals and considerable variability in effect sizes, though meta-regression was conducted to address this. Furthermore, differences in trial designs introduce limitations in cross-trial comparisons.

## Conclusions

Anti-Aβ mAb therapy slows cognitive decline, but with small effect sizes, and raises potential concerns about ARIA and headaches.

## Author summary

### Why was this study done?

- Alzheimer's disease (AD) is a major societal challenge that creates a significant burden on patients, families, and healthcare systems worldwide.

- Abnormal deposits of amyloid-beta (Aβ) in the brain are a hallmark of AD pathology.

- Monoclonal antibodies (mAbs) targeting Ab are a new treatment option aimed at slowing cognitive decline in patients with AD.

- Although several mAbs have been tested in clinical trials, there has been limited comprehensive review of their overall safety and effectiveness.

### What did the researchers do and find?

- We analyzed 13 randomized controlled trials to assess the efficacy and safety of anti-Ab mAb therapy for AD.

- It was revealed that anti-Aβ mAb therapy slows cognitive decline with small effect sizes, but elevated risk of amyloid-related imaging abnormalities with edema/hemorrhage (ARIA-E/ARIA-H) and headache were noted.

- Early use of anti-Aβ mAbs appears to be more beneficial in slowing cognitive decline.

### What do these findings mean?

- Clinicians should fully understand the risks and benefits of anti-Aβ mAb therapies and discuss these in detail with patients and their families.

- Given that cognitive decline can complicate the understanding of their risk/benefit, it is advisable to start consultations on the implementation of mAb therapy as soon as patients are diagnosed with AD while also considering both medical and socioeconomic factors.

- One limitation of the study is that the trials were designed differently, making it hard to directly compare their results. Also, we did not have detailed data on individual patients.

## Introduction

Alzheimer's disease (AD) is a major societal challenge that poses a considerable global burden on patients, families, and healthcare systems. Abnormal amyloid-beta (Aβ) deposition in the brain is a key pathological hallmark of AD and one of the major targets in AD research and drug development [1]. Recently, the amyloid hypothesis was tested in phase III randomized controlled trials (RCTs) with several potent monoclonal antibodies (mAbs) that target Aβ peptides [2–4]. These trials resulted in traditional approval of donanemab and lecanemab by the U.S. Food and Drug Administration (FDA) and conditional approval of aducanumab [5,6]. In January 2024, the manufacturer of aducanumab discontinued its commercialization [7], leaving lecanemab and donanemab as the only commercially available anti-Aβ mAbs when this review was conducted. The main concern surrounding these drugs relates to patient safety relative to clinical efficacy. These safety concerns include amyloid-related imaging abnormalities (ARIAs), which consist of cerebral edema (ARIA-E) and cerebral microhemorrhage and hemosiderosis (ARIA-H). As other rare side effects, such as death, do not show significant differences in individual RCTs due to insufficient power, meta-analysis could be useful in this context. Although it is anticipated that more detailed treatment outcomes in the real-world setting will be available through Phase IV trials or observational studies, meta-analyses of existing studies can provide useful and immediate insights to guide patient care and drug development. Although some meta-analyses have been previously published [8–12], these have not included recently published RCTs regarding donanemab and gantenerumab [4,13]. Thus, there is a growing interest in meta-regression analyses to address the impact of using different antibodies on efficacy and safety profiles. In this context, we conducted an updated systematic review including meta-regression analyses to investigate the efficacy and safety of mAbs against Aβ for AD.

## Methods

### Search strategy and selection criteria

Preferred Reporting Items for Systematic review and Meta-Analysis (PRISMA) guidelines were followed (**Appendix A in** S1 File). We searched for published trials on Pubmed, Embase, and ClinicalTrials.gov on January 14th, 2024 with a combination of the following keywords: "Alzheimer", "mild cognitive impairment", "dementia", "antibody", "passive immunotherapy", "randomized controlled trial, or "randomized clinical trial." Detailed criteria for searching in each database are given in **Text A in** S1 File. Study selection, review, and data extraction were performed by two authors independently, and any disagreements were resolved by consensus.

To assess the quality of trial reports we used the Jadad scale, which focuses on the study's randomization, blinding, and descriptions of withdrawals and dropouts [14]. To be included in the meta-analysis, a study had to: (1) Be a phase III randomized, placebo-controlled blinded trial that tested the efficacy and safety of mAbs against Aβ with parallel design; (2) Have a Jadad score >3 [14]; (3) Include participants with mild cognitive impairment (MCI) or any stage of dementia due to AD; and (4) Include at least 200 participants for cognitive and safety

measures in each group. We included studies that had been published in any language. A study was excluded if it: (1) Was a narrative/systematic review, meta-analysis, phase I, II, or IV trial, a single-arm trial, case-control study, prospective cohort study, cross-sectional study, case report, opinion/editorial, trial design paper, or post-hoc or secondary analysis of a main study; (2) Involved familial AD or other dementias; (3) Did not report cognitive/functional outcomes (e.g., studies that only reported imaging or biomarker results); (4) Was an open-label trial; or (5) Included fewer than 200 patient cognitive and safety outcomes in each arm. The screening process was conducted between January 15th 2024 and February 14th 2024.

## Outcomes and modifiers

Primary outcomes (end-points) included cognitive measures reported by the majority of the original studies: Mean change from baseline in Clinical Dementia Rating scale-Sum of Boxes (CDR-SB) and AD Assessment Scale-Cognitive Subscale (ADAS-Cog). Secondary outcomes were total events of death, serious adverse events (SAEs), ARIA-E, ARIA-H, headaches, falls, and dizziness. SAE was defined according to the individual definitions provided by each study and generally encompassed death, life-threatening circumstances, required inpatient hospitalization or prolongation of existing hospitalization, or persistent disability/incapacity. Total events of ARIA-E were defined as reported events of ARIA-E occurring at any level of severity, ranging from mild abnormalities detected on brain imaging, which do not always cause serious clinical symptoms, to more severe ARIA-E with serious consequences. ARIA-H included events reported as ARIA-H and microhemorrhages identified in MRI images taken as part of a predetermined procedure regardless of symptoms. However, studies that reported data exclusively on microhemorrhages necessitating treatment, without including numbers for those that were asymptomatic, were not included in the analysis for ARIA-H. The tertiary exploratory endpoint was intraparenchymal cerebral macrohemorrhage. This condition involved significant hemorrhaging within the brain's parenchyma, characterized by deformation, destruction, or swelling of brain tissues exceeding the scale of microhemorrhages.

Modifiers for meta-regression analyses were selected based on one of the following criteria: (1) Their reported association with AD progression; (2) The hypothesis that they could influence the effects of amyloid-beta antibody therapy and were thus considered in the trial's patient inclusion criteria; and (3) Characteristics of the antibody that suggest these factors could impact treatment outcomes. As a result, the following were selected as modifiers: Age, baseline cognitive scores on the mini-mental state examination (MMSE), percentage of *APOE4* carriers included in trials, Aβ burden on positron emission tomography (PET), AD stage (classified as "early AD," which includes only MCI or mild AD, and "mild-moderate," which includes mild and moderate AD according to the revised National Institute on Aging-Alzheimer's Association (NIA-AA) criteria [15]), drug, antibody type (human versus humanized), binding mechanisms of the antibody (monomer-specific, oligomer- or aggregates-binding, or plaque-specific [binds to a pyroglutamate form of Aβ]), and biological effectiveness (measured by the effect of decreasing Aβ burden in the brain via amyloid PET). While we attempted to standardize the definitions of moderators as much as possible, there were inherent differences in antibody characteristics and patient profiles across studies. For example, for AD stage, among the trials classified as "early", EXPEDITION 3 (which tested solanezumab), only included patients with mild AD but not MCI [16]. Regarding binding mechanisms, solanezumab was classified as monomer-targeting, donanemab as plaque-specific, while oligomer- or aggregate-binding antibodies exhibit varying affinities to fibrils, oligomers, and monomers [2,4,16–18]. Furthermore, baseline Aβ burden in the brain, measured by amyloid PET and reported in centiloids, was available only in sub-study populations

within a limited subset of trials. Consequently, meta-regression analyses for baseline Aβ burden were conducted using these sub-studies.

Changes in Aβ or tau levels using PET, as well as Aβ or phosphorylated-tau (p-tau) levels in cerebral spinal fluid (CSF) or plasma, were reported. Although these data were not included in the pooled analysis, they have been collected and summarized.

## Data analysis

Publication bias was assessed using Egger's test and Funnel's plot for each outcome reported in at least 10 trials. A *P*-value of < 0.05 for Egger's test was considered statistically significant. To assess heterogeneity, $\tau^2$ and $I^2$ statistics were reported and Cochran's *Q* test was performed, where a *P*-value of < 0.05 was considered statistically significant.

Heterogeneity was classified as mild when $I^2$ was less than 30%, moderate when $I^2$ ranged from 30% to 50%, and substantial when $I^2$ exceeded 50% [19]. $\tau^2$ was calculated using the Mandel–Paule algorithm to estimate between-study variance [20]. The analysis was conducted on the intention-to-treat population. Cases with multiple dosage groups in antibody arms were combined and referred to as the "antibody therapy group" (detailed in **Texts B** and **C in S1 File**) [21,22]. The summary measure of CDR-SB was expressed as mean difference (MD), while standardized mean difference (SMD) was expressed for ADAS-Cog, as it was reported in either ADAS-Cog version 11, 13, or 14 in each trial. The calculation methods for the SMD are described in **Text D in** S1 File [23]. SMD $\leqq$ 0.2 was considered small, $\leqq$ 0.5 moderate, and 0.8 $\leqq$ large. Summary measures for binary outcomes were expressed as a risk ratio (RR). For pairwise meta-analyses, we reported results based on both the common-effect model and the random-effect model. The common-effect model assumes that all studies estimate the same true effect, while the random-effects model accounts for between-study variation by assuming that each study may have a different true effect and combines these results to estimate an overall average effect. Results of subgroup analyses on drugs were also reported for primary endpoints. To estimate the range into which the effects of future studies could be expected to fall based on present evidence, we also reported prediction intervals (PIs) [24]. In order to assess the clinical relevance of the statistically significant results, we computed the "Number Needed to Treat" (NNT) or the "Number Needed to Harm" (NNH) [25,26]. The detailed methodology for the calculation of NNT is described in **Text E in** S1 File. When data were available from 10 or more trials, meta-regression was performed to evaluate the impact of modifiers. The meta-regression was performed under the assumption that the modifiers had a linear relationship with the outcome and that the heterogeneity observed could be partly explained by these modifiers. Mixed-effects meta-regression models were applied, with each analysis including one modifier [27].

For sensitivity analyses, we first excluded antibodies without any biological effect on reducing Aβ burden in the brain or CSF. Next, we excluded trials with a maximum antibody dosage below 3 mg/kg. Third, we excluded trials halted due to pre-specified interim analysis. Subsequently, we excluded trials from prior sensitivity analyses. Then, we included only the lowest or highest dosage group and the placebo group of each trial. Finally, we included trials previously excluded due to having fewer than 200 participants per arm, to assess their impact on the robustness of the findings. Statistical analyses were performed using the meta package [28,29] version 7.0-0 in R version 4.3.1 (R Foundation for Statistical Computing, Vienna, Austria). To ensure the consistency of results, we also performed comparative analyses for selected examples using the metafor package version 4.6-0 [30,31], focusing on both categorical and continuous outcomes, as well as meta-regression. Representative case studies are presented, and detailed comparisons are provided in **Text F in** S1 File.

## Results

The literature search identified 13 phase III RCTs, which included 18,826 patients with MCI or dementia due to AD (**Fig 1**). Baseline characteristics of the patients and trials are shown in **Tables 1** and **2**.

Two studies tested gantenerumab [13], one study tested donanemab [4], one study tested lecanemab [2], two studies tested aducanumab [3], three studies tested solanezumab [16,17], and four studies tested bapineuzumab [32,33]. The follow-up periods generally ranged between 7 and 80 weeks, while the GRADUATE I and II trials that investigated gantenerumab were 116 weeks. In all studies, concomitant standard AD medications such as acetylcholine esterase inhibitors or memantine were allowed, alone or in combination. In the trials for

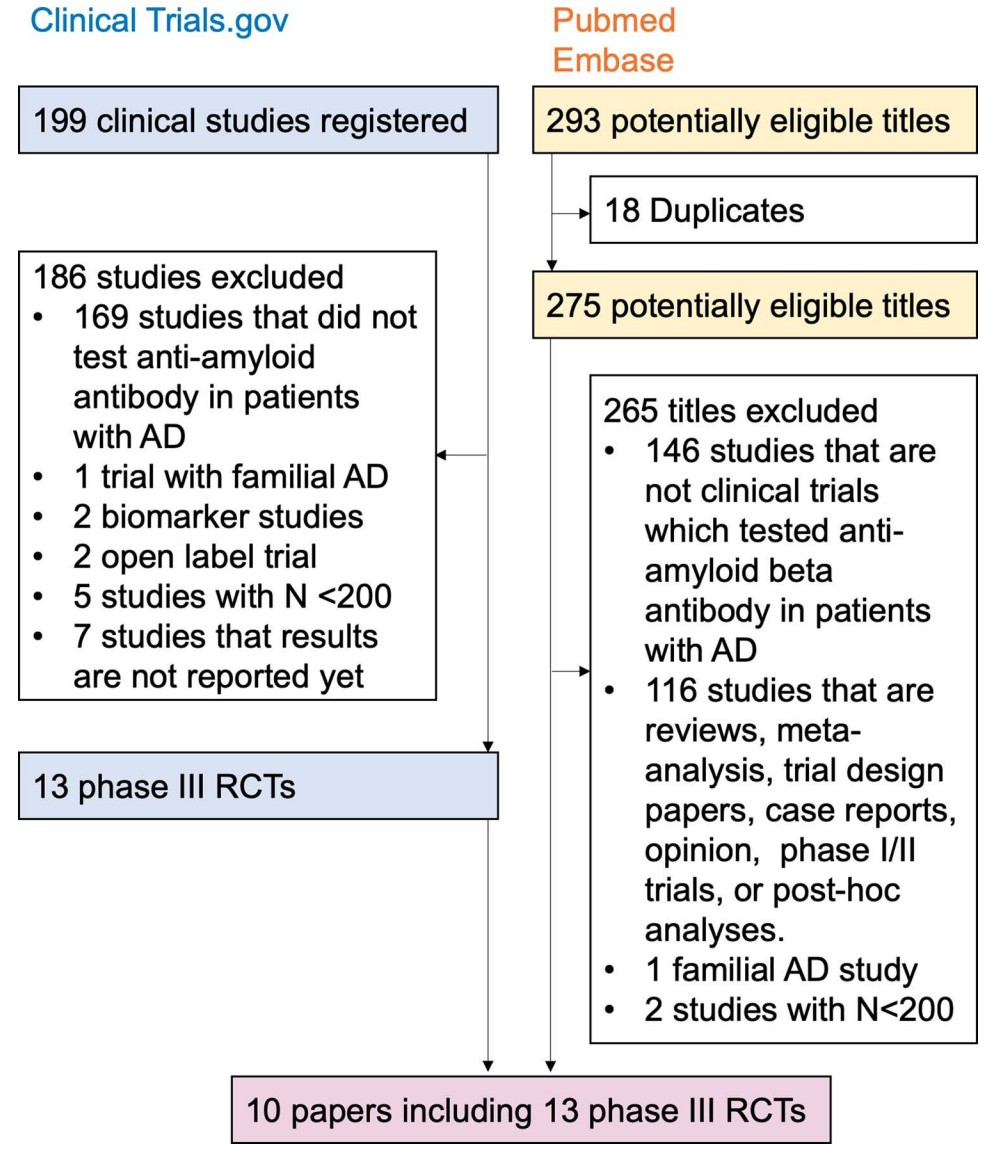

**Fig 1. Flow chart of study selection process.** AD, Alzheimer's disease; RCT, randomized controlled trial.

**Table 1. Patient characteristics of included trials.**

| | No. patients | | Female (%) | | Age (years) | | Baseline MMSE | | ApoE4 carriage (%) | | Baseline Ab burden in PET (centiloids) | |
|---|---|---|---|---|---|---|---|---|---|---|---|---|
| Trial name | A | P | A | P | A | P | A | P | A | P | A | P |
| GRADUATE I [13] | 499 | 485 | 58.1 | 52.6 | 71.1 (7.9) | 72.1 (7.8) | 23.5 (3.3) | 23.6 (3.0) | 65.3 | 67.6 | 99.4 | 96.1 |
| GRADUATE II [13] | 498 | 477 | 57.8 | 59.7 | 71.6 (7.8) | 71.8 (7.4) | 23.6 (3.1) | 23.8 (3.2) | 66.9 | 67.3 | 95.6 | 90.7 |
| TRAILBLAZER-ALZ2 [4] | 794 | 838 | 57.3 | 57.4 | 73.0 (6.2) | 73.0 (6.2) | 22.4 (3.8) | 22.2 (3.9) | 69.8 | 71.2 | 103.5 | 101.6 |
| Clarity_AD [2] | 859 | 875 | 51.6 | 53.0 | 71.4 (7.9) | 71.0 (7.8) | 25.5 (2.2) | 25.6 (2.2) | 68.9 | 68.6 | 77.9 | 75.0 |
| EMERGE (low dose) [3] | 543 | 548 | 49.5 | 53.0 | 70.6 (7.4) | 70.8 (7.4) | 26.3 (1.7) | 26.4 (1.8) | 66.7 | 67.2 | 88.0*2 | 83.6*2 |
| EMERGE (high dose) [3] | 547 | 548 | 51.9 | 53.0 | 70.6 (7.5) | 70.8 (7.4) | 26.3 (1.7) | 26.4 (1.8) | 66.7 | 67.2 | 85.4*2 | 88.0*2 |
| ENGAGE (low dose) [3] | 547 | 545 | 51.9 | 52.6 | 70.4 (7.0) | 69.8 (7.7) | 26.4 (1.8) | 26.4 (1.7) | 71.5 | 69.0 | 85.9*2 | 83.8*2 |
| ENGAGE (high dose) [3] | 555 | 545 | 52.6 | 52.6 | 70.0 (7.7) | 69.8 (7.7) | 26.4 (1.8) | 26.4 (1.7) | 68.2 | 69.0 | 90.9*2 | 83.8*2 |
| EXPEDITION 3 [16] | 1,057 | 1,072 | 56.8 | 58.6 | 72.7 (7.8) | 73.3 (8.0) | 22.8 (2.8) | 22.6 (2.9) | 69.3 | 66.3 | – | – |
| EXPEDITION 1 [17] | 506 | 506 | 59.1 | 56.7 | 75.0 (7.9) | 74.4 (8.0) | 21(4) | 21(3) | 57.3 | 61.3 | – | – |
| EXPEDITION 2 [17] | 521 | 519 | 54.3 | 55.1 | 72.5 (8.0) | 72.4 (7.8) | 21(3) | 21(3) | 56.8 | 59.5 | – | – |
| Study 3000 (low dose) [32] | 255 | 328 | 71.1 | 69.7 | 71.1*1 | 69.7*1 | 20.8 (3.2) | 20.8 (3.1) | 0.0 | 0.0 | – | – |
| Study 3000 (high dose) [32] | 253 | 328 | 70.7 | 69.7 | 70.7*1 | 69.7*1 | 20.8 (3.1) | 20.8 (3.1) | 0.0 | 0.0 | – | – |
| Study 3001 [32] | 650 | 431 | 64.5 | 60.1 | 70.9*1 | 70.2*1 | 20.9 (3.1) | 21.0 (3.0) | 100.0 | 100.0 | – | – |
| Study 301 (low dose) [33] | 314 | 493 | 52.5 | 50.3 | 73.1 (9.3) | 71.9 (10.1) | 21.2 (3.4) | 21.2 (3.2) | 0.0 | 0.0 | – | – |
| Study 301 (high dose) [33] | 307 | 493 | 57.0 | 50.3 | 73.5 (9.1) | 71.9 (10.1) | 21.2 (3.3) | 21.2 (3.2) | 0.0 | 0.0 | – | – |
| Study 302 [33] | 658 | 432 | 54.4 | 56.0 | 72.0 (8.0) | 72.3 (9.5) | 20.8 (3.1) | 20.7 (3.2) | 100.0 | 100.0 | – | – |

Age, CDR-SB score, and MMSE scores were shown as mean (SD). A: antibody, P: Placebo, CDR-SB: Clinical Dementia Rating-Sum of Boxes, MMSE: Mini-mental state examination.

*1 SD was not reported in the Study 3000 and Study 3001.

*2 For EMERGE/ENGAGE trials, baseline uptake of amyloid beta was reported using standardized uptake value ratio (SUVR). The authors provided a conversion equation, which was used to convert the SUVR values into centiloids.

aducanumab [3] and bapineuzumab [32,33], there were two dosage groups for the antibody, so these were combined to one antibody group.

## Risk of bias

Funnel plots for primary endpoints are shown in S1 Fig. Egger's test *P*-values were 0.04 for CDR-SB and 0.75 for ADAS-Cog, respectively. Trials with small sample size that favored antibody therapy were potentially missing in CDR-SB. There was no evidence of asymmetry in funnel plots in secondary endpoints (S2 Fig).

## Primary outcomes

Changes in CDR-SB and ADAS-Cog were reported in all trials. Compared with placebo, treatment with mAbs significantly improved cognitive performance on CDR-SB (random effect model: MD −0.25, 95% CI [−0.38, −0.11]), and ADAS-Cog (random effect model: SMD −0.09, 95% CI [−0.12, −0.06]), in which a negative change indicates an improvement for both scores (**Figs 2 and 3**). Heterogeneity was moderate for CDR-SB ($\tau^2 = 0.03$, $I^2 = 53.10\%$, $p = 0.01$) and low for ADAS-Cog ($\tau^2 = 0.00$, $I^2 = 6.97\%$, $p = 0.84$). The effect size for ADAS-Cog was small (the absolute value of the SMD < 0.2). The results of meta-regression analyses showed that a lower baseline MMSE was associated with a smaller SMD in ADAS-Cog ($p = 0.02$) (**Table 3 and S4 Fig**), but no association was found between baseline MMSE and change in CDR-SB (**Table 3 and S3 Fig**).

**Table 2. Characteristics of the trials and anti-amyloid beta antibodies used in trials.**

| Trial name | Antibody | Year | Antibody type | AD stage | Binding Mechanism | Biological effect | Primary efficacy outcome | Duration w: weeks m: months |
|---|---|---|---|---|---|---|---|---|
| GRADUATE I [13] | Gantenerumab | 2023 | Human | Early | Oligomers or aggregates | Yes | CDR-SB | 116 w |
| GRADUATE II [13] | Gantenerumab | 2023 | Human | Early | Oligomers or aggregates | Yes | CDR-SB | 116 w |
| TRAILBLAZER-ALZ2 [4] | Donanemab | 2023 | Humanized | Early | Plaque only*1 | Yes | iADRS | 76 w |
| Clarity_AD [2] | Lecanemab | 2023 | Humanized | Early | Oligomers or aggregates | Yes | CDR-SB | 18 m |
| EMERGE (2 doses) [3] | Aducanumab | 2019 | Human | Early | Oligomers or aggregates | Yes | CDR-SB | 78 w |
| ENGAGE (2 doses) [3] | Aducanumab | 2019 | Human | Early | Oligomers or aggregates | Yes | CDR-SB | 78 w |
| EXPEDITION 3 [16] | Solanezumab | 2018 | Humanized | Early*2 | Monomers | No | ADAS-Cog 14 | 80 w |
| EXPEDITION 1 [17] | Solanezumab | 2014 | Humanized | Mild-moderate | Monomers | No | ADAS-Cog 11 | 80 w |
| EXPEDITION 2 [17] | Solanezumab | 2014 | Humanized | Mild-moderate | Monomers | No | ADAS-Cog 11 | 80 w |
| Study 3000 (2 doses) [32] | Bapineuzumab | 2016 | Humanized | Mild-moderate | Oligomers or aggregates | Yes | ADAS-Cog 11 | 78 w |
| Study 3001 [32] | Bapineuzumab | 2016 | Humanized | Mild-moderate | Oligomers or aggregates | Yes | ADAS-Cog 11 | 78 w |
| Study 301 (2 doses) [33] | Bapineuzumab | 2014 | Humanized | Mild-moderate | Oligomers or aggregates | Yes | ADAS-Cog 11 | 78 w |
| Study 302 [33] | Bapineuzumab | 2014 | Humanized | Mild-moderate | Oligomers or aggregates | Yes | ADAS-Cog 11 | 78 w |

*1 Donanemab binds to a pyroglutamate form of Amyloid beta, which is closely related to plaque formation.

*2 In EXPEDITION 3, only patients with mild AD were included, and no MCI patients were recruited.

AD, Alzheimer's Disease; CDR-SB, Clinical Dementia Rating-Sum of Boxes; ADAS-Cog, Alzheimer's Disease Assessment Scale-Cognitive Subscale.

There was no statistically significant association between age, percentage of *APOE4* carrier included in trials, or baseline Aβ burden measured by PET and change in CDR-SB or ADAS-Cog. Trials involving patients with early-stage AD were associated with a significant negative impact on the effect size for both CDR-SB ($p = 0.02$) and ADAS-Cog ($p = 0.01$, **Table 3 and Tables E and F in** S1 File). Also, compared to lecanemab, use of bapineuzumab was associated with a significant negative impact on the CDR-SB ($p = 0.03$) and ADAS-Cog ($p = 0.01$). Furthermore, when CDR-SB was used as a metric for cognitive function, the trials that used antibodies that bind to a pyroglutamate form of Aβ, which aggregates in amyloid plaque, demonstrated a significant beneficial effect in reducing cognitive decline when compared to trials using antibodies that target Aβ monomers ($p = 0.03$).

## Secondary outcomes (safety outcomes)

The information available from each study is summarized in **Table A in** S1 File. In Expeditions 1 and 2 [17], the pooled results for safety outcomes were included in the analyses since the separate results were not available. The results of meta-analysis and meta-regression analysis on safety outcomes are reported in **Figs 4**, **5**, **and S7–S20 and Table 4 and** Table B in S1 File.

Death was reported in all trials, and no difference was found in the risk of death between the antibody and placebo groups (**Fig 4A**). The definition of SAEs was similar across trials (**Text G in** S1 File) and it was available in 11 trials except for Study 3000 and 3001 [32], in which it was not clearly defined. Overall, no significant association was observed between the use of antibodies and the risk of SAEs (S7 Fig). ARIA-E was reported in all 13 trials and mAb therapies were associated with increased risk of ARIA-E, but heterogeneity was high (random-effect model: RR 9.79, 95% CI [5.32,18.01], $\tau^2 = 0.87$, $I^2 = 59.91\%$, $p < 0.01$, **Fig 4B**). There was a statistically significant interaction between the binding mechanism of the antibody and the risk of ARIA-E, indicating a higher risk of observing ARIA-E in trials that used antibodies that preferentially bind to Aβ oligomers/aggregates ($p = 0.01$) or plaque only

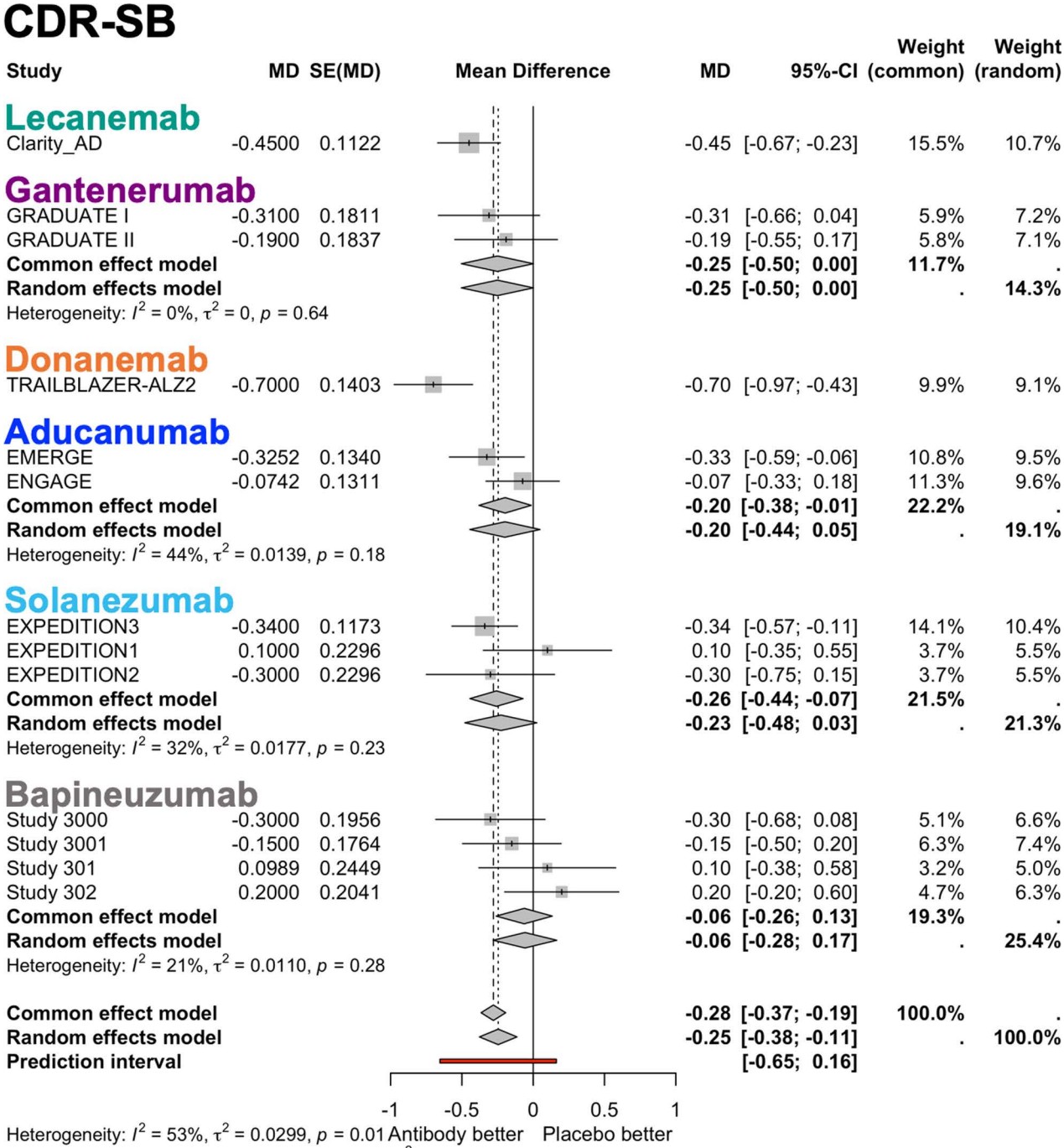

**Fig 2. Forest plots for change in Clinical Dementia Rating-Sum of Boxes (CDR-SB).** The summary measure of CDR-SB was expressed as Mean difference (MD), and the standard errors (SE) and 95% confidence interval (CI) are presented. The overall results are shown for both the random effects Model and the common effect model, along with the results of the subgroup analysis by antibody. Forest plots show the effect size and its 95% CI, with the size of the gray squares representing the weight of each study. $I^2$ value, $\tau^2$ value calculated using the Mandel–Paule algorithm, and the $P$-value from Cochran's $Q$ test for heterogeneity are provided. The $\chi^2$ value, degrees of freedom (df), and $P$-values from the test for subgroup differences are also reported.

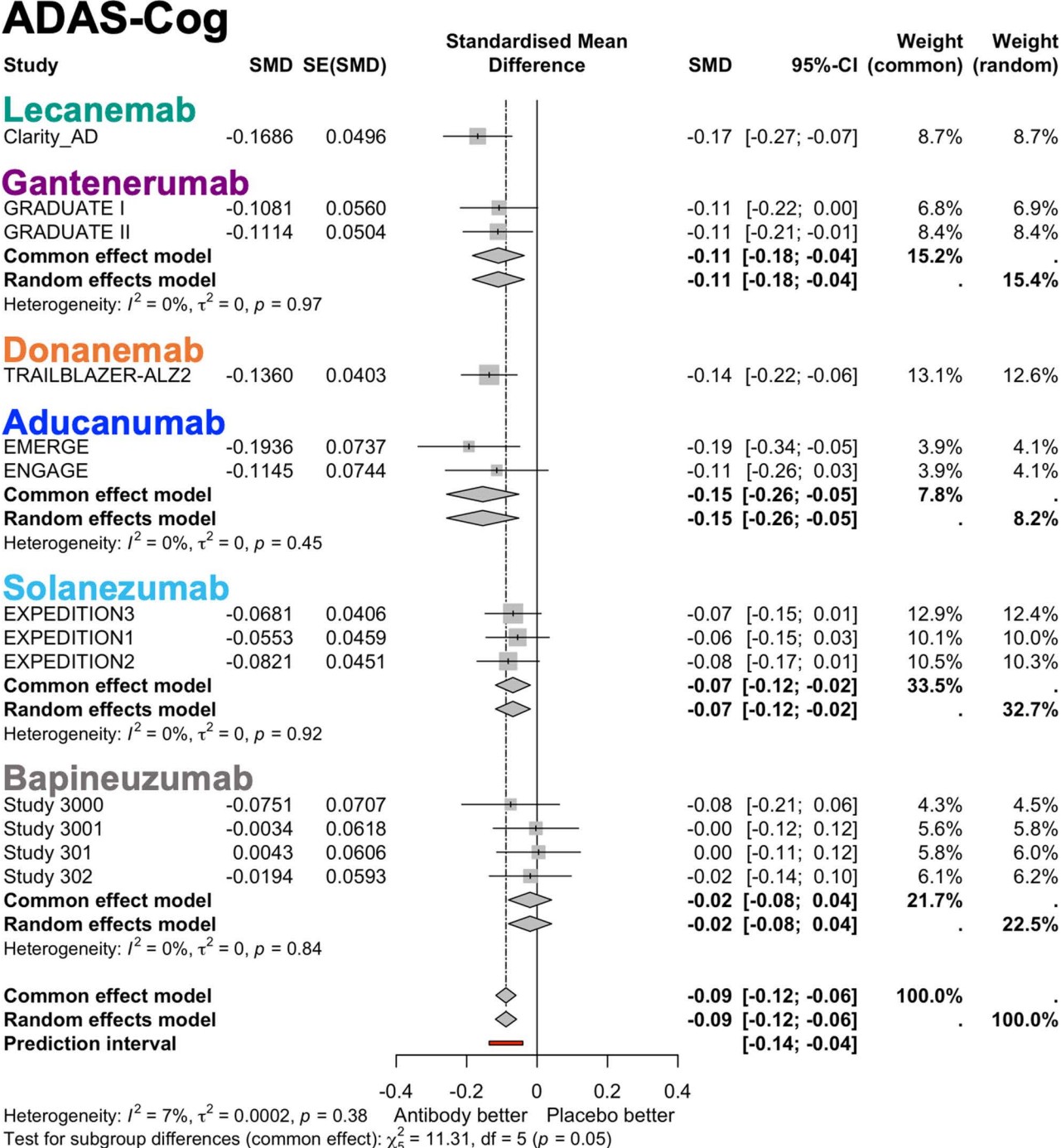

**Fig 3. Forest plots for change in Alzheimer's Disease Assessment Scale-Cognitive Subscale (ADAS-Cog).** The summary measure of ADAS-Cog was expressed as standardized mean difference (SMD), with the standard errors (SE) and 95% confidence interval (CI) presented for each study. The overall results are shown for both the random effects model and the common effect model, along with the results of the subgroup analysis by antibody. Forest plots show the effect size and its 95% CI, with the size of the gray squares representing the weight of each study. $I^2$ value, $\tau^2$ value calculated using the Mandel–paule algorithm, and the $P$ value from Cochran's $Q$ test for heterogeneity are provided. The $\chi^2$ value, degrees of freedom (df), and $P$-values from the test for subgroup differences are also reported.

**Table 3. Results of meta-regression analyses for efficacy endpoints.**

|  | CDR-SB | | ADAS-Cog | |
|---|---|---|---|---|
|  | MD (95% CI) | *P*-value | SMD (95% CI) | *P*-value |
| **Mean age** | −0.05 (−0.16, 0.05) | 0.33 | −0.003 (−0.02, 0.03) | 0.82 |
| **Mean MMSE** | −0.03 (−0.10, 0.04) | 0.39 | −0.02 (−0.04, −0.003) | 0.02* |
| **APOE4 carrier percentage** | −0.0001 (−0.01, 0.01) | 0.98 | 0.000 (−0.002, 0.001) | 0.66 |
| **Aβ burden on PET** | −0.01 (−0.03, 0.01) | 0.54 | 0.002 (−0.003, 0.01) | 0.48 |
| **AD-Stage** | MD (95% CI) from placebo | P-values for comparison with the reference group | SMD (95% CI) from placebo | *P*-values for comparison with the reference group |
| Early (reference) | −0.35 (−0.49, −0.21) | NA | −0.12 (−0.16, −0.08) | NA |
| Mild-moderate | −0.07 (−0.26, 0.12) | 0.02* | −0.04 (−0.09, 0.00) | 0.01* |
| **Drug** | MD (95% CI) from placebo | *P*-values for comparison with the reference group | SMD (95% CI) from placebo | *P*-values for comparison with the reference group |
| Lecanemab (reference) | −0.45 (−0.73, −0.17) | NA | −0.17 (−0.27, −0.07) | NA |
| Gantenerumab | −0.25 (−0.53, 0.03) | 0.33 | −0.11 (−0.18, −0.04) | 0.35 |
| Donanemab | −0.70 (−1.03, −0.37) | 0.26 | −0.14 (−0.22, −0.06) | 0.61 |
| Aducanumab | −0.20 (−0.42, 0.02) | 0.17 | −0.15 (−0.26, −0.05) | 0.84 |
| Solanezumab | −0.24 (−0.46, −0.02) | 0.25 | −0.07 (−0.12, −0.02) | 0.07 |
| Bapineuzumab | −0.06 (−0.28, 0.16) | 0.03* | −0.02 (−0.08, 0.04) | 0.01* |
| **Antibody type** | MD (95% CI) from placebo | *P*-values for comparison with the reference group | SMD (95% CI) from placebo | *P*-values for comparison with the reference group |
| human (reference) | −0.22 (−0.46, 0.02) | NA | −0.13 (−0.18, −0.07) | NA |
| Humanized | −0.25 (−0.42, −0.09) | 0.83 | −0.08 (−0.11, −0.05) | 0.17 |
| **Binding mechanism** | MD (95% CI) from placebo | *P*-values for comparison with the reference group | SMD (95% CI) from placebo | *P*-values for comparison with the reference group |
| Monomer (reference) | −0.23 (−0.47, 0.01) | NA | −0.07 (−0.12, −0.02) | NA |
| Oligomer or Aggregates | −0.20 (−0.34, −0.07) | 0.85 | −0.09 (−0.13, −0.05) | 0.55 |
| Plaque only | −0.70 (−1.06, −0.34) | 0.03* | −0.14 (−0.22, −0.05) | 0.19 |
| **Biological effect** | MD (95% CI) from placebo | *P*-values for comparison with the reference group | SMD (95% CI) from placebo | *P*-values for comparison with the reference group |
| Yes (reference) | −0.25 (−0.41, 0.10) | NA | −0.10 (−0.13, −0.06) | NA |
| No | −0.21 (−0.51, 0.08) | 0.83 | −0.07 (−0.12, −0.02) | 0.38 |

Meta-regression was performed only when the information from 10 or more trials were available. For the categorical modifiers (AD-stage, drug, antibody type, binding mechanism and biological effect), mean difference (MD) or standardized mean difference (SMD) from placebo, and *P*_values for comparison with the reference group are reported. Reference groups were shown in the table.

*$P$-value < 0.05.

CI, confidence interval; AD, Alzheimer's disease; Aβ, amyloid beta; PET, positron emission tomography; CDR-SB, Clinical Dementia Rating-Sum of Boxes; ADAS-Cog, Alzheimer's Disease Assessment Scale-Cognitive Subscale; MD, mean difference; NA, not applicable; SMD, standardized mean difference; MMSE, mini-mental state examination.

($p = 0.02$) compared to those that bind Aβ monomers (**Table 4** and S14A Fig). Also, antibodies with no biological effect were associated with a lower risk of observing ARIA-E ($p = 0.003$) (S14B Fig). ARIA-H was reported in only eight trials, and mAb therapies were associated with increased risk of ARIA-H with high heterogeneity (random-effect model: RR 1.94, 95% CI [1.47,2.57], $\tau^2 = 0.12$, $I^2 = 78.85\%$, $p < 0.01$) (**Fig 4C**). Headache and fall were reported in 11 trials, whereas dizziness was reported in 10 trials. MAb therapies were associated with an increased risk of headache (random-effect model: RR 1.21, 95% CI [1.10,1.32], $\tau^2 = 0.00$, $I^2 = 0.00\%$, $p = 0.47$), but no difference was found in the risk of fall and dizziness (**Fig 5**). Heterogeneity was low for headache, fall, and dizziness. There was a statistically significant

**Table 4. Results of meta-regression analyses for serious adverse events, ARIA-E, and headache.**

| | Serious adverse event | | ARIA-E | | Headache | |
|---|---|---|---|---|---|---|
| | **Risk Ratio (95% CI)** | *P*-value | **Risk Ratio (95% CI)** | *P*-value | **Risk Ratio (95% CI)** | *P*-value |
| **Mean age** | 0.97 (0.91, 1.04) | 0.41 | 0.67 (0.46, 0.98) | 0.04* | 0.96 (0.90, 1.03) | 0.25 |
| **Mean MMSE** | 0.99 (0.94, 1.04) | 0.65 | 0.95 (0.67, 1.33) | 0.75 | 1.04 (1.00, 1.09) | 0.07 |
| **APOE4 carrier percentage** | 1.00 (1.00, 1.01) | 0.65 | 1.00 (0.97, 1.02) | 0.89 | 1.00 (1.00, 1.01) | 0.42 |
| **Aβ burden on PET** | 0.99 (0.98, 1.01) | 0.33 | 1.01 (0.98, 1.04) | 0.37 | 1.00 (0.99, 1.02) | 0.83 |
| **AD-Stage** | **RR compared to placebo (95% CI)** | ***P*-values for comparison with the reference** | **RR compared to placebo (95% CI)** | ***P*-values for comparison with the reference** | **RR compared to placebo (95% CI)** | ***P*-values for comparison with the reference** |
| Early (reference) | 0.98 (0.86, 1.10) | NA | 8.32 (3.78, 18.30) | NA | 1.27 (1.15, 1.41) | NA |
| Mild-moderate | 1.10 (0.94, 1.30) | 0.24 | 13.25 (4.56, 38.52) | 0.49 | 1.01 (0.83, 1.22) | 0.04* |
| **Drug** | **RR compared to placebo (95% CI)** | ***P*-values for comparison with the reference** | **RR compared to placebo (95% CI)** | ***P*-values for comparison with the reference** | **RR compared to placebo (95% CI)** | ***P*-values for comparison with the reference** |
| Lecanemab (reference) | 1.25 (0.97, 1.60) | NA | 7.52 (1.82, 31.09) | NA | 1.37 (1.03, 1.82) | NA |
| Gantenerumab | 0.82 (0.67, 1.02) | 0.01* | 9.56 (3.45, 26.55) | 0.79 | 1.27 (0.98, 1.63) | 0.69 |
| Donanemab | 1.10 (0.88, 1.37) | 0.46 | 11.67 (2.83, 47.22) | 0.67 | 1.42 (1.09, 1.84) | 0.86 |
| Aducanumab | 0.98 (0.81, 1.18) | 0.13 | 11.43 (4.20, 31.06) | 0.64 | 1.29 (1.10, 1.52) | 0.73 |
| Solanezumab | 0.94 (0.84, 1.06) | 0.05 | 1.45 (0.33, 6.39) | 0.12 | 0.97 (0.79, 1.19) | 0.06 |
| Bapineuzumab | 1.19 (1.01, 1.40) | 0.76 | 19.33 (7.40, 50.51) | 0.28 | 1.06 (0.83, 1.35) | 0.18 |
| **Antibody type** | **RR compared to placebo (95% CI)** | ***P*-values for comparison with the reference** | **RR compared to placebo (95% CI)** | ***P*-values for comparison with the reference** | **RR compared to placebo (95% CI)** | ***P*-values for comparison with the reference** |
| human (reference) | 0.91 (0.77, 1.07) | NA | 10.53 (3.72, 29.81) | NA | 1.28 (1.12, 1.47) | NA |
| humanized | 1.07 (0.96, 1.20) | 0.10 | 9.36 (4.02, 21.80) | 0.86 | 1.15 (1.01, 1.30) | 0.23 |
| **Binding mechanism** | **RR compared to placebo (95% CI)** | ***P*-values for comparison with the reference** | **RR compared to placebo (95% CI)** | ***P*-values for comparison with the reference** | **RR compared to placebo (95% CI)** | ***P*-values for comparison with the reference** |
| Monomer (reference) | 0.94 (0.76, 1.16) | NA | 1.51 (0.39, 5.82) | NA | 0.97 (0.79, 1.19) | NA |
| Oligomer or Aggregates | 1.04 (0.91, 1.19) | 0.41 | 11.51 (7.35, 18.02) | 0.01* | 1.25 (1.12, 1.39) | 0.04* |
| Plaque only | 1.10 (0.79, 1.52) | 0.42 | 11.67 (3.71, 36.69) | 0.02* | 1.42 (1.09, 1.84) | 0.03* |
| **Biological effect** | **RR compared to placebo (95% CI)** | ***P*-values for comparison with the reference** | **RR compared to placebo (95% CI)** | ***P*-values for comparison with the reference** | **RR compared to placebo (95% CI)** | ***P*-values for comparison with the reference** |
| Yes (reference) | 1.05 (0.93, 1.18) | NA | 11.23 (7.75, 16.3) | NA | 1.27 (1.15, 1.41) | NA |
| No | 0.94 (0.77, 1.14) | 0.32 | 1.55 (0.43, 5.54) | 0.003* | 0.97 (0.79, 1.19) | 0.02* |

For the categorical modifiers (AD-stage, drug, antibody type, binding mechanism and biological effect), risk ratio compared to placebo and P-values for comparison with the reference group are shown. Reference groups are shown in the table.

*P-value < 0.05.

MMSE, mini-mental state examination; CI, confidence interval; AD, Alzheimer's Disease; Aβ, amyloid-beta; PET, positron emission tomography; NA, not applicable; RR, risk ratio.

## (a) Death

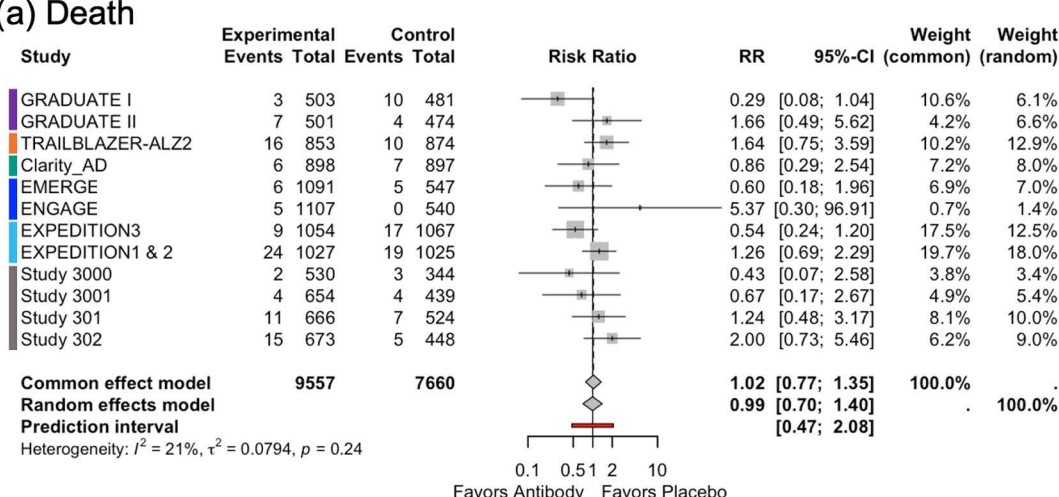

## (b) ARIA-E

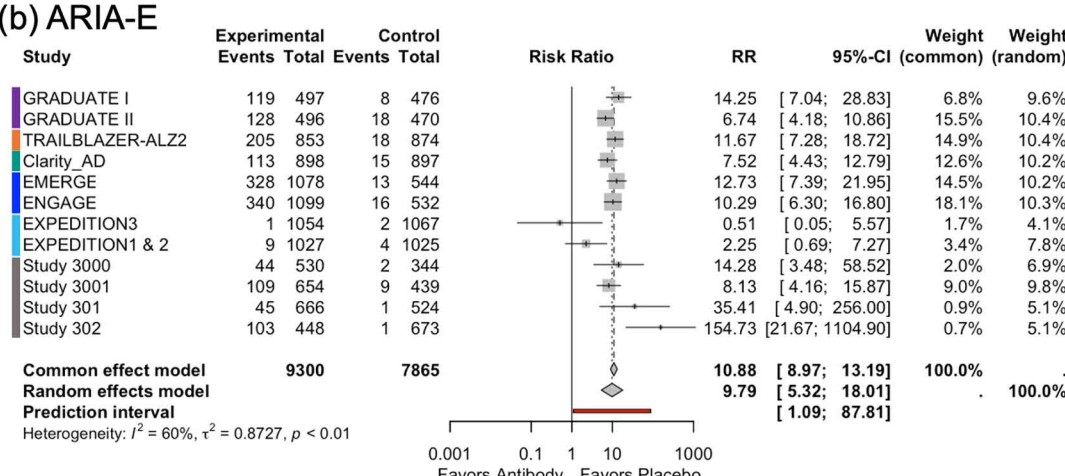

## (c) ARIA-H

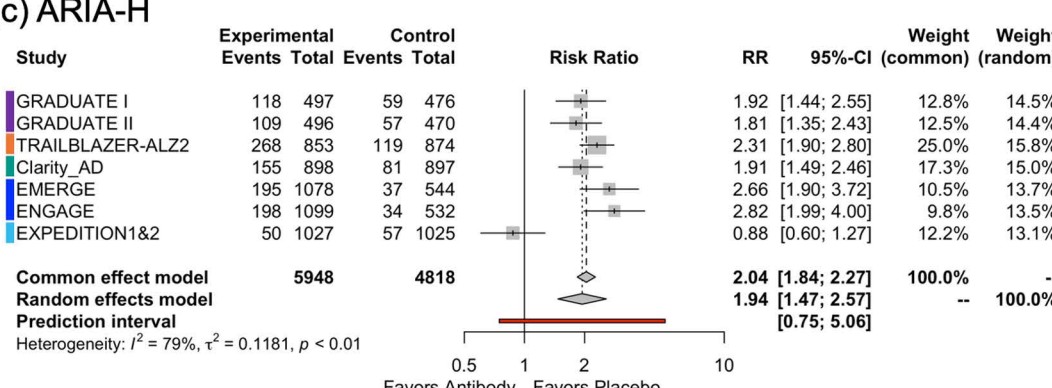

| Gantenerumab | Donanemab | Lecanemab | Aducanumab | Solanezumab | Beprineuzumab |

**Fig 4. Forest plots for safety outcomes.** (**a**) Death, (**b**)ARIA-E, and (**c**) ARIA-H. A number of events in experimental and control groups, risk ratio (RR), and its 95% confidence interval (CI), and weights are shown. The overall results are shown for both the random effects model and the common effect model. Forest plots show the effect size and its 95% CI, with the size of the gray squares representing the weight of each study. $I^2$ value, $\tau^2$ value calculated using the Mandel–Paule

algorithm, and the *P*-value from Cochran's *Q* test for heterogeneity are provided. Additionally, the prediction interval is also presented. ARIA-E, Amyloid-Related Imaging Abnormalities-Effusion; ARIA-H, Amyloid-Related Imaging Abnormalities-Hemorrhage.

interaction between AD stage and the risk of headache ($p = 0.04$), indicating a greater risk of headache in trials including patients with mild to moderate AD compared to trials including patients with early-stage AD (**Table 4** and S16A Fig). Also, antibodies with no biological effect were associated with a lower risk of headache ($p = 0.02$, S16E Fig).

## Tertiary outcome

Although there was some variation in the definition of cerebral macrohemorrhage (**Text H in** S1 File), we included 8 trials in the analysis. In the random-effect model, the association between mAb therapies and risk of cerebral macrohemorrhage was not significant (RR 1.77, 95% CI [0.88,3.57]), though there was a significant association observed in the common-effect model (RR 1.84, 95% CI [1.01,3.36]). Heterogeneity was relatively mild ($\tau^2 = 0.09$, $I^2 = 14.71\%$, $p = 0.32$) (S21 Fig).

## Number needed to treat/harm

The NNT was 8 for CDR-SB and 4 for ADAS-Cog (**Table E in** S1 File). For adverse events, NNH was 7 for ARIA-E, 10 for ARIA-H, and 54 for headache.

## Sensitivity analyses

In total, we performed eight sensitivity analyses and summarized the result in **Table 5**.

The results of the sensitivity analyses were consistent with the main analyses in both primary and secondary outcomes. First, we performed sensitivity analyses, excluding antibodies that did not show any biological effect of decreasing Aβ burden in the brain, which was solanezumab [16,17]. The results were similar to the main analyses (S22–S24 Figs). Second, we omitted trials where the maximum antibody dosage was < 3 mg/kg, assuming an average patient weight of 100 kg. This excluded the trials with bapineuzumab [32,33], as the dosage in these studies ranged only from 0.5 to 1.0 mg/kg. The resultant effect sizes for primary endpoints were slightly larger than the main analyses (CDR-SB: MD 0.31, 95% CI [−0.46, −0.17]; ADAS-Cog: SMD −0.10, 95% CI [−0.14, −0.08]) (S25–S27 Figs). Third, we excluded the trials that halted early as a result of the interim analysis, EMERGE, and ENGAGE, which evaluated aducanumab. The resultant effect sizes for primary and secondary endpoints were similar to the main analysis (S28–S30 Figs). Moreover, we omitted both solanezumab and bapineuzumab, which were omitted in the first and second sensitivity analyses. This caused the effect sizes for primary endpoints to increase further (CDR-SB: MD −0.35, 95% CI [−0.53, −0.17]; ADAS-Cog: SMD −0.14, 95% CI [−0.18, −0.09]) (S31 Fig). Nonetheless, the absolute value of the SMD for ADAS-Cog remained < 0.2, indicating a small effect on improving cognition. Consistent with the primary analysis, the risk of ARIA-E and headache were higher in the antibody-treated group (S32 **and** S33 Fig). Subsequently, we omitted trials with solanezumab, bapineuzumab, and aducanumab, which were omitted in the first, second, and third sensitivity analyses (S34–S36 Figs). This caused the effect sizes for primary endpoints, especially the change in CDR-SB, to increase further (CDR-SB: MD −0.44, 95% CI [−0.65, −0.23]; ADAS-Cog: SMD −0.13, 95% CI [−0.18, −0.09]).

In the main analysis, the groups with two doses of the antibody arm were combined to form a single antibody group, but analyses were also conducted separately for both the

## (a) Headache

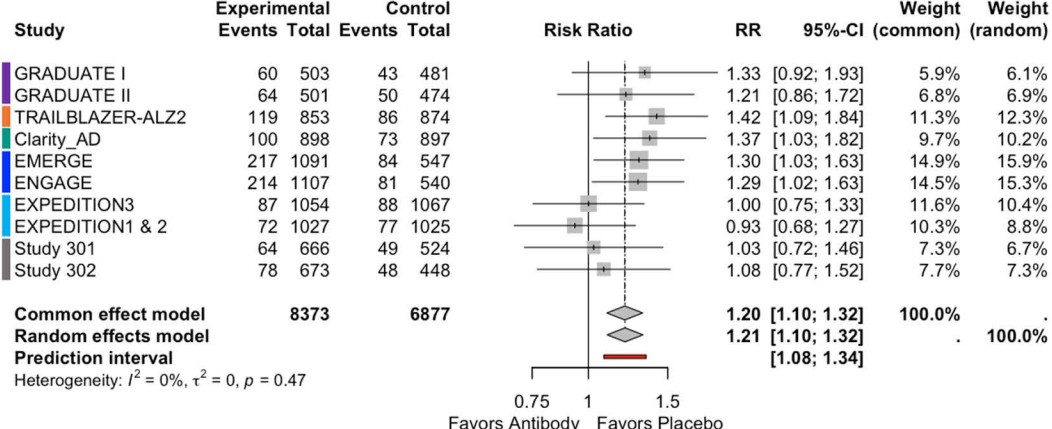

## (b) Fall

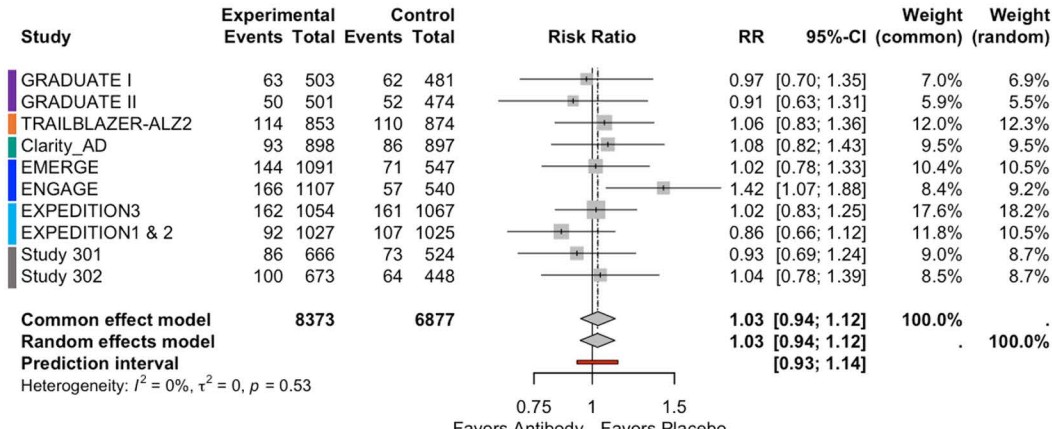

## (c) Dizziness

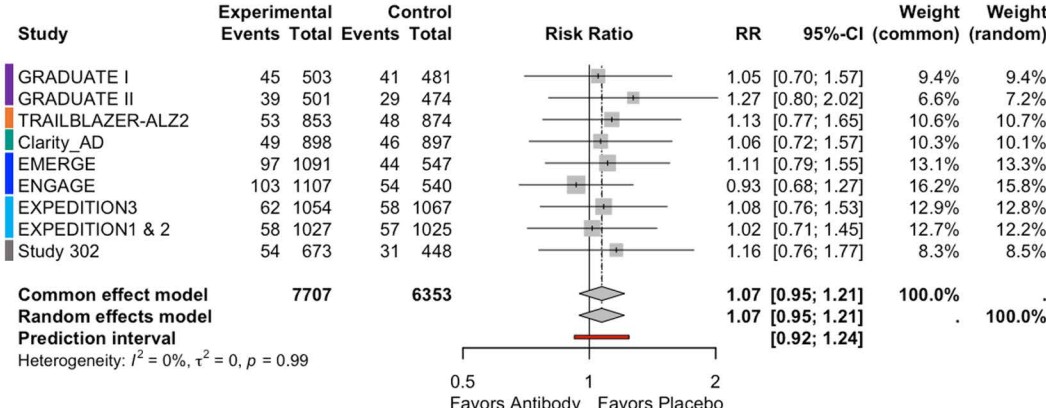

**Fig 5. Forest plots for safety outcomes related to symptoms.** (**a**) Headache, (**b**) Fall, and (**c**) Dizziness. The overall results are shown for both the random effects model and the common effect model. Forest plots show the risk ratio (RR) and its 95% confidence interval (CI), with the size of the gray squares representing the weight of each study. $I^2$ value, $\tau^2$ value calculated using the Mandel–Paule algorithm, and the *P*-value from Cochran's *Q* test for heterogeneity are provided. Additionally, the prediction interval is also presented.

**Table 5. Summary of sensitivity analyses.**

| | CDR-SB | ADAS-Cog | Death | SAE | ARIA-E | ARIA-H | Headache | Fall | Dizziness |
|---|---|---|---|---|---|---|---|---|---|
| | MD (95% CI) | SMD (95% CI) | RR (95% CI) | RR (95% CI) | RR (95% CI) | RR (95% CI) | RR (95% CI) | RR (95% CI) | RR (95% CI) |
| Main analysis | −0.25 (−0.38, −0.11) | −0.09 (−0.12, −0.06) | 0.99 (0.70, 1.40) | 1.02 (0.92, 1.13) | 9.79 (5.32, 18.01) | 1.94 (1.47, 2.57) | 1.21 (1.10, 1.32) | 1.03 (0.94, 1.12) | 1.07 (0.95, 1.21) |
| Excluding solanezumab | −0.25 (−0.41, −0.09) | −0.10 (−0.14, −0.06) | 1.04 (0.69, 1.58) | 1.05 (0.93, 1.19) | 11.2 (7.74, 16.31) | 2.17 (1.89, 2.50) | 1.27 (1.15, 1.41) | 1.06 (0.95, 1.17) | 1.08 (0.93, 1.24) |
| Excluding bapineuzumab | −0.31 (−0.46, −0.17) | −0.11 (−0.14, −0.08) | 0.94 (0.58, 1.48) | 0.98 (0.88, 1.08) | 7.72 (4.39, 13.57) | 1.94 (1.47, 2.57)[*1] | 1.23 (1.11, 1.36) | 1.04 (0.94, 1.14) | 1.06 (0.93, 1.21) |
| Excluding aducanumab | −0.25 (−0.41, −0.10) | −0.08 (−0.11, −0.05) | 1.01 (0.70, 1.45) | 1.03 (0.91, 1.16) | 9.47 (4.30, 20.89) | 1.71 (1.25, 2.36) | 1.17 (1.04, 1.31) | 0.99 (0.90, 1.09) | 1.10 (0.95, 1.27) |
| Exclusing solanezumab and bapineuzumab | −0.35 (−0.53, −0.17) | −0.14 (−0.18, −0.09) | 0.97 (0.52, 1.84) | 1.00 (0.87, 1.15) | 9.85 (7.81, 12.42) | 2.17 (1.89, 2.50)[*2] | 1.32 (1.18, 1.48) | 1.08 (0.96, 1.22) | 1.07 (0.92, 1.24) |
| Excluding solanezumab, bapineuzumab and aducanumab | −0.44 (−0.65, −0.23) | −0.13 (−0.18, −0.09) | 0.97 (0.45, 2.11) | 1.00 (0.81, 1.24) | 9.23 (6.61, 12.88) | 2.04 (1.81, 2.31) | 1.35 (1.15, 1.57) | 1.02 (0.88, 1.18) | 1.12 (0.91, 1.37) |
| Included only the low-dose group | −0.26 (−0.39, −0.14) | −0.09 (−0.12, −0.06) | 0.96 (0.63, 1.45) | 1.01 (0.92, 1.12) | 8.88 (5.00, 15.77) | 1.89 (1.46, 2.44) | 1.19 (1.08, 1.31) | 1.01 (0.93, 1.11) | 1.06 (0.93, 1.20) |
| Included only the high-dose group | −0.23 (−0.37, −0.08) | −0.08 (−0.12, −0.05) | 1.08 (0.77, 1.50) | 1.02 (0.92, 1.13) | 10.5 (5.51, 20.11) | 1.99 (1.47, 2.69) | 1.22 (1.11, 1.34) | 1.03 (0.94, 1.13) | 1.10 (0.96, 1.25) |
| Including trials with patients < 200 per arm in addition to the trials in the main analysis | −0.22 (−0.35, −0.10) | −0.09 (−0.12, −0.06) | 0.89 (0.92,1.09) | 1.00 (0.92, 1.09) | 9.09 (5.26, 15.69) | 1.71 (1.33, 2.20) | 1.16 (1.05, 1.27) | 1.03 (0.95, 1.11) | 1.08 (0.96, 1.21) |

[*1]ARIA-H was not reported in bapineuzumab trials; therefore the results are the same as that in the main analysis.

[*2]ARIA-H was not reported in bapineuzumab trials; therefore the results are the same as that in the sensitivity analysis that excluded only solanezumab.

CDR-SB, Clinical Dementia Rating-Sum of Boxes; ADAS-Cog, Alzheimer's Disease Assessment Scale-Cognitive Subscale; SAE, serious adverse event; ARIA-E, amyloid-related imaging abnormalities with edema or effusion; ARIA-H, amyloid-related imaging abnormalities with hemorrhage; MD, mean difference; SMD, standardized mean difference; RR, risk ratio.

low-dose group and the high-dose group. In analyses that included low- or high-dose groups only, similar to the main analyses, mAb therapies were associated with slower progression of cognitive impairment measured with CDR-SB and ADAS-Cog (S37 **and** S41 Fig), with increasing risk of ARIA-E, ARIA-H, and headache and no change of risk in death, SAEs, fall, dizziness, and cerebral macrohemorrhage (S38–S40, S42 **and** S43 Fig).

Finally, we included trials excluded in the main analysis due to having fewer than 200 participants per arm, to assess their impact on the robustness of the findings [34,35]. All three trials were halted early due to futility. Two trials tested crenezumab but the efficacy endpoint results were available only in CREAD and not in CREAD2 [34]. Funnel plots and P-values for Egger's test for primary and secondary endpoints are shown in S44–S46 Figs. Egger's test P-values were 0.04 for CDR-SB. Consistent with the main analysis, trials with small sample size that favored antibody therapy were potentially missing in CDR-SB. There was no evidence of asymmetry in funnel plots in secondary endpoints. The resultant effect sizes for primary endpoints were slightly smaller than the main analyses in CDR-SB, and similar in ADAS-Cog (CDR-SB: MD 0.22, 95% CI [−0.35, −0.10]; ADAS-Cog: SMD −0.09, 95% CI [−0.12, −0.06]) (S47–S48 Figs). Results for secondary endpoints were consistent to the main analysis (S49–S51 Figs).

### Neuroimaging and biomarker results

The results of Aβ and tau PET are summarized in **Tables C and D in** S1 File, respectively. Amyloid PET was conducted as a sub-study in all trials. Significant reductions in brain Aβ uptake were observed in the experimental groups in trials involving gantenerumab,

donanemab, lecanemab, and aducanumab, while no difference was noted in trials using solanezumab. In the bapineuzumab studies, only Study 302 showed a decrease in Aβ burden. Tau PET results were reported only in recent trials with small sample sizes. In the pooled results from EMERGE and ENGAGE (aducanumab), tau burden decreased in regions such as the medial temporal lobe in the experimental arm, but no significant changes were seen in other regions. No difference in tau burden between the treatment and the placebo group was observed in trials with gantenerumab and donanemab, and results from the Clarity-AD trial are yet to be reported.

CSF and plasma biomarker results are summarized in **Tables E and F in** S1 File, respectively. There was significant variation in the types of biomarkers measured across the trials. The level of Aβ$_{42}$, which is associated with Alzheimer's pathology due to its tendency to aggregate more readily than other forms of Aβ [36], was reported to be increased in the CSF in most of the trials. Plasma Aβ levels were reported only in a few trials. Interestingly, it was reported in several trials that plasma and CSF p-tau levels significantly decreased in the treatment group.

## Discussion

Here, we have updated and expanded the findings of previous meta-analyses by including recently reported donanemab and gantenerumab studies. Various meta-regression analyses were also conducted to explore differences due to the characteristics of patients included in the trials and the antibodies used.

The findings of our updated meta-analyses on the efficacy and safety of anti-Aβ mAb therapies in patients with AD can be summarized as follows: (1) Anti-Aβ mAb therapies are associated with slowing cognitive decline compared with placebo; however, the overall effect is small; (2) Anti-Aβ mAb therapies are associated with increased risk of ARIA-E, ARIA-H, and headache compared with placebo; (3) No association was observed between anti-Aβ mAb therapies and risk of death, SAEs, fall, dizziness, or cerebral macrohemorrhage; (4) Trials including patients in the early stages of AD yielded more favorable results than those involving patients with mild-to-moderate AD; (5) Compared to lecanemab, bapineuzumab may be less effective; (6) No significant increase in the risk of side effects was observed between lecanemab and other antibodies.

The focus of our screening strategy and selection process was large RCTs with at least 200 patients in each arm in efficacy/safety outcomes, similar to the previous report [8]. We also included the latest trial results for donanemab and gantenerumab [4,13]. This strategy contributed to the robustness of our results but also resulted in exclusion of some insightful trials which were terminated early due to futility, such as phase III trials of crenezumab (NCT03114657, NCT02670083) [34] or another phase III trial of gantenerumab (NCT 01224106) [35]. The Egger's test *P*-value was < 0.05 for CDR-SB change, suggesting an absence of smaller-size trials with favorable outcomes for mAb therapies. Notably, even when three trials with fewer than 200 participants per group were included, the Egger's test *P*-value for CDR-SB change remained < 0.05, as these trials were terminated early due to lack of efficacy [34,35]. Despite this, the risk of publication bias remains low. This is because this meta-analysis exclusively includes RCTs, which are typically pre-registered on platforms such as ClinicalTrials.gov. This pre-registration process for RCTs implies a reduced risk of publication bias in our analysis.

In terms of the efficacy, no trial demonstrated that mAb halted cognitive decline or improved cognitive function in patients with AD relative to baseline measurement. Substantial heterogeneity was observed in the change in CDR-SB scores, whereas heterogeneity was

mild when assessed by ADAS-Cog. In CDR-SB, the criteria include aspects such as attendance in community affairs, personal care, and home and hobbies. Significant individual differences in the timing of losing these abilities might have led to greater variability across trials. On the other hand, ADAS-Cog focuses on more detailed cognitive abilities, which may allow for a more sensitive assessment of cognitive maintenance or decline. The mild heterogeneity observed with ADAS-Cog suggests that this measure may provide a more reliable assessment of the overall trend in cognitive decline compared to CDR-SB.

Although statistically significant, the effect sizes associated with slowing cognitive decline were small. The MD between the mAb treatments and placebo was −0.25 on the 18-point CDR-SB score, which is smaller than the commonly accepted thresholds of 0.50 for clinically meaningful change in patients with MCI [37,38]. Also, the absolute value of the SMD was less than 0.2 when using ADAS-Cog as a measure, indicating a similarly small effect. Even after the exclusion of solanezumab and bapineuzumab, which either lacked a biological effect in reducing brain Aβ burden or were administered at significantly lower doses, the SMD between the remaining antibodies and placebo remained at −0.12, indicating a small effect. Importantly, sensitivity analysis, including smaller studies with fewer than 200 participants per arm, showed consistent findings in ADAS-Cog, which confirms the robustness of our results. These results are consistent with previous findings for the same drug class [8–13]. To explore the source of heterogeneity, we performed multiple meta-regression analyses. Compared to lecanemab, bapineuzumab was less effective in slowing cognitive decline, likely due to its significantly lower doses, ranging from three to 10 times lower than those used in trials of other drugs. This indicates that the observed lack of efficacy might not stem from bapineuzumab's inherent properties, but rather from insufficient administration. Reflecting the positive trials focused on patients with early AD, such as those with lecanemab and donanemab, we observed a significant interaction between the recruitment of patients with early AD and the slowing of cognitive decline. However, the impact on later-stage patients remains unclear for lecanemab and donanemab due to the lack of trials, and are thus awaiting future real-world data.

For safety endpoints, mAb therapy targeting Aβ was associated with an increased risk of ARIA-E, ARIA-H, and headache. ARIA-E and ARIA-H demonstrated substantial heterogeneity, SAE exhibited moderate heterogeneity, and other safety endpoints showed mild heterogeneity across trials. Substantial heterogeneity in ARIA was mainly driven by solanezumab, which uniquely targets monomers and has not demonstrated a biological effect in reducing brain Aβ burden. The lower ARIA risk in solanezumab might be related to its preferential binding to Aβ monomers, which may be cleared from the brain without affecting aggregated Aβ. The consequences of ARIA were reported to be rarely severe enough to meet the definition of SAE, and mAb use was not associated with risk of SAEs in our analysis. In the TRAILBLAZER-ALZ2 trial that investigated donanemab, three ARIA-associated deaths were reported, indicating that ARIA should not be underestimated [4]. The use of mAbs was also associated with an increased risk of experiencing headache. Similar to the risk of observing ARIA-E, the use of antibodies that target monomers was inversely associated with the risk of headache. This result is consistent with the reports indicating that headache is one of the symptoms associated with ARIA [39]. The risk stratification for individuals with ARIA who experience more severe outcomes or headaches, along with management strategies and long-term consequences of ARIA, remain unclear and thus await further phase IV trial study and real-world data. Although the definition of cerebral macrohemorrhage differed across trials, we analyzed the association between mAb use and cerebral macrohemorrhage. In the random effect model, although there was a trend associated with mAb use, no statistically significant association was found. However, in the open-label lecanemab study, three deaths from brain hemorrhage were reported, suggesting that close patient monitoring is warranted [40].

NNT was 8 for difference in CDR-SB mean change compared with placebo, indicating a small effect size. And NNHs were 7,10, and 54 for ARIA-E, ARIA-H, and Headache, respectively. One thing to be cautious about is that although Kraemer's method [25] for calculation of NNT for continuous outcomes is used in several meta-analyses [8,41], the definitions differ slightly from the conventional definition of NNT, and as a result, the calculated NNT is slightly lower than the actual observed NNTs [42]. Therefore, the true NNT for continuous outcomes might be slightly higher.

We did not perform a meta-analysis for neuroimaging and fluid biopsy results due to (1) Variability in the radiotracers used in PET scans and measured biomarkers, (2) PET results being reported as standardized uptake value ratios (SUVR), which complicating direct comparisons, and (3) The absence of Aβ levels in CSF or plasma from the TRAILBLAZER-ALZ2 trial of donanemab, and excluding this data could lead to misleading conclusions. In addition, raw neuroimaging and fluid biopsy data were not fully released by these phase III trials yet. lecanemab, gantenerumab, and aducanumab showed a reduction in brain Aβ burden as measured by amyloid PET following treatment, alongside increases in CSF Aβ42 levels. However, despite the reduction in brain Aβ burden, the impact on cognitive decline has been small in this study, reflecting the complex nature of AD. Other factors, such as neuroinflammation, oxidative tissue damage, or mitochondrial dysfunction also significantly contribute to disease progression in AD [36]. Addressing these factors simultaneously may be crucial for slowing cognitive decline effectively. Notably, these trials reported decreased p-tau levels in both CSF and plasma, potentially supporting the amyloid cascade hypothesis by suggesting that Aβ accumulation modulates other AD pathologies, such as tau [36]. These findings are based on a limited number of patients, and further investigation is needed to better understand these effects.

When considering the application of mAb therapy, it is crucial to consider not only the drug's effect on the patient but also the broader societal impact, such as reduced caregiver burden. Reports on the cost-effectiveness of commercially available anti-Aβ antibodies conducted in the US suggest that, when compared to standard care without the use of these antibodies, lecanemab and donanemab extend quality-adjusted life years by 0.61 and 0.62 years, respectively [43,44]. While the results of medico-economic effects analysis depend heavily on regional variations in caregiving costs, which calls for caution in interpretation, the price of lecanemab, 26,500 USD per year, is considered to be set lower than its anticipated socioeconomic value in the US [43]. A study that analyzed the cost-effectiveness of donanemab, assuming an annual price of 28,000 USD, which is lower than the actual 32,000 USD per year, found that donanemab would not be cost-effective at this price point [44]. However, donanemab's unique dosing regimen, where treatment is paused once significant amyloid reduction is achieved, suggests that anti-amyloid antibody treatments could still be cost-effective if they are sufficiently effective. In contrast, in the United Kingdom, National Institute for Health and Care Excellence did not recommend the use of either lecanemab or donanemab within the national health service due to their limited clinical benefits and high cost. [45,46]. Thus, the cost-effectiveness of mAb therapy remains controversial, with significant variations in assessment across countries. Further observation is necessary to verify the cost-effectiveness of these antibodies in real-world settings.

We acknowledge several potential limitations. First, ADAS-Cog scores were reported using different versions (11, 13, and 14), and SMD were calculated for standardization. However, no imputation was performed for unique items in ADAS-Cog 13 and 14, relying on the assumption that these items correlate with shared items across versions, which is a limitation. Reports on secondary or tertiary endpoints were also unavailable in some RCTs, and no imputation was performed for these cases. Second, due to slight variations in the definition of cerebral

macrohemorrhage across trials, results must be interpreted with caution. Moreover, statistical heterogeneity was relatively high for ARIA-E and ARIA-H, which produced wide CIs and indicates that the true effect sizes in these cases varies considerably. To address this heterogeneity, meta-regression analyses were conducted. However, analyses concerning patient-level characteristics were not performed. It is also important to note that trial designs and analysis methods differed slightly across trials, and we could not take every difference into account. For example, the trial that tested donanemab was stopped in patients who reached low levels of amyloid and had shortened exposures, which might have reduced the risk of side effects compared with other antibodies. Also, in the same trial that tested donanemab, the presence of phosphorylated tau 181 was part of the eligibility criteria, which was later removed through a protocol amendment. Finally, careful interpretation of NNT is required, as it may sometimes be calculated lower than the actual observed NNT [25,42].

In conclusion, this updated meta-analysis provides evidence that mAb therapies for AD significantly slow cognitive decline compared with placebo, though the effect size remains small. Some antibodies were able to slow cognitive deterioration, while none of the antibodies demonstrated an improvement from baseline cognitive function. These antibody therapies were associated with higher risk of ARIA-E, ARIA-H and headache. Particularly regarding ARIA, there is insufficient knowledge about the stratification of risks that could lead to more severe outcomes, making further investigation necessary. The long-term results of this treatment remain unclear; drugs appear to be more effective when started early. Since cognitive decline complicates the understanding of its risk-benefit, it is advisable to start consultations on its use as soon as patients are diagnosed with AD, considering both medical and socioeconomic factors.

## Supporting information

**S1 File. Supplemental Materials. Appendix A.** Preferred Reporting Items for Systematic review and Meta-Analysis Protocols (PRISMA-P) 2015 checklist: recommended items to address in a systematic review protocol. **Text A.** Search strategy. **Text B.** Data extraction for continuous outcomes. **Text C.** Methodologies to combine groups. **Text D.** Calculation methods for the standard mean difference (SMD). **Text E.** Calculation of number needed to treat/harm (NNT)/(NNH). **Text F.** Comparison between meta and metafor packages. **Text G.** Definition of serious adverse events (SAE). **Text H.** Definition of cerebral macrohemorrhage. **Table A.** Trials that were included in the analyses. **Table B.** Results of meta-regression analyses for death, fall, and dizziness. **Table C.** Summary of amyloid PET and Tau PET results. **Table D.** Summary of biomarker results. **Table E.** Number needed to treat/harm.
(DOCX)

**S1 Fig. Funnel plots for efficacy endpoints.** (**a**) The Clinical Dementia Rating-Sum of Boxes (CDR-SB) and (**b**) Alzheimer's Disease Assessment Scale-Cognitive Subscale (ADAS-Cog). Filled circles represent estimated treatment effect (risk ratio) and its precision (standard error) for each individual study. In addition to individual study results, the fixed effect estimate (vertical dashed line) with 95% confidence interval limits (diagonal dashed lines) and the random effects estimate (vertical dotted line) are shown in the figures. Also, *P*-values of Egger's test are shown. *$P$-value < 0.05.
(PDF)

**S2 Fig. Funnel plots for safety endpoints.** (**a**) Death, (**b**) Serious Adverse Event, (**c**) ARIA-E, (**d**) Headache, and (**e**) Fall. Filled circles represent estimated treatment effect (risk ratio) and its precision (standard error) for each individual study. In addition to individual study results,

the fixed effect estimate (vertical dashed line) with 95% confidence interval limits (diagonal dashed lines) and the random effects estimate (vertical dotted line) are shown in the figures. Also, *P*-values of Egger's test are shown. \*$P$-value < 0.05. ARIA-E, Amyloid-Related Imaging Abnormalities-Effusion; ARIA-H, Amyloid-Related Imaging Abnormalities-Hemorrhage.
(PDF)

**S3 Fig. Bubble plots showing the results of meta-regression of CDR-SB mean difference by (a) Mean age, (b) MMSE score, (c) ApoE4 carrier percentage, and (d) Amyloid beta burden in PET.** The size of the bubbles shows the inverse of the variance of the mean difference in each trial, with larger bubbles indicating trials with higher precision. The *P*-values from the meta-regression analysis are also reported. CDR-SB, the Clinical Dementia Rating-Sum of Boxes; MMSE, Mini-Mental State Examination; PET, positron emission tomography.
(PDF)

**S4 Fig. Bubble plot showing the results of meta-regression of ADAS-Cog standardized mean difference by difference by (a) Mean age, (b) MMSE score, (c) ApoE4 carrier percentage, and (d) Amyloid beta burden in PET.** The size of the bubbles shows the inverse of the variance of the standardized mean difference in each trial, with larger bubbles indicating trials with higher precision. The p-values for comparison with the reference group (shown as "ref") from the meta-regression analysis are also reported on the top of the bubble plots. \*$P$-value < 0.05. ADAS-Cog, Alzheimer's Disease Assessment Scale-Cognitive Subscale; MMSE, Mini-Mental State Examination; PET, positron emission tomography.
(PDF)

**S5 Fig. Bubble plots showing the results of meta-regression of CDR-SB standardized mean difference by (a) AD stage, (b) Drug, (c) Antibody type, (d) Binding mechanism, and (e) Biological effect.** The size of the bubbles shows the inverse of the variance of the mean difference in each trial, with larger bubbles indicating trials with higher precision. The *p*-values from the meta-regression analysis are also reported. \*$P$-value < 0.05. CDR-SB, the Clinical Dementia Rating-Sum of Boxes; AD, Alzheimer's Disease.
(PDF)

**S6 Fig. Bubble plots showing the results of meta-regression of ADASA-Cog standardized mean difference by (a) AD stage, (b) Drug, (c) Antibody type, (d) Binding mechanism, and (e) Biological effect.** The size of the bubbles shows the inverse of the variance of the standardized mean difference in each trial, with larger bubbles indicating trials with higher precision. The *p*-values for comparison with the reference group (shown as "ref") from the meta-regression analysis are also reported on the top of the bubble plots. \*$P$-value < 0.05. ADAS-Cog, Alzheimer's Disease Assessment Scale-Cognitive Subscale; AD, Alzheimer's Disease.
(PDF)

**S7 Fig. Forest plot showing the results of meta-analysis for serious adverse events.**
(PDF)

**S8 Fig. Bubble plots showing the results of meta-regression of death, by (a) Mean age, (b) MMSE score, (c) ApoE4 carrier percentage, and (d) Amyloid beta burden in PET.** The size of the bubbles shows the inverse of the variance of the log-transformed risk ratio in each trial, with larger bubbles indicating trials with higher precision. The *p*-values from the meta-regression analysis are also reported. MMSE, Mini-Mental State Examination; PET, Positron Emission Tomography.
(PDF)

**S9 Fig. Bubble plot showing the results of meta-regression of death, by (a) AD stage, (b) Drug, (c) Antibody type, (d) Binding mechanism, and (e) Biological effect.** The size of the bubbles shows the inverse of the variance of the log-transformed risk ratio in each trial, with larger bubbles indicating trials with higher precision. The *p*-values for comparison with the reference group (shown as "ref") from the meta-regression analysis are also reported on the top of the bubble plots. AD, Alzheimer's Disease.
(PDF)

**S10 Fig. Bubble plot showing the results of meta-regression of occurrence of serious adverse event, by (a) Mean age, (b) MMSE score, (c) ApoE4 carrier percentage, and (d) Amyloid beta burden in PET.** The size of the bubbles shows the inverse of the variance of the log-transformed risk ratio in each trial, with larger bubbles indicating trials with higher precision. The *p*-values from the meta-regression analysis are also reported. MMSE, Mini-Mental State Examination; PET, positron emission tomography.
(PDF)

**S11 Fig. Bubble plot showing the results of meta-regression of occurrence of serious adverse event, by (a) AD stage, (b) Drug, (c) Antibody type, (d) Binding mechanism, and (e) Biological effect.** The size of the bubbles shows the inverse of the variance of the log-transformed risk ratio in each trial, with larger bubbles indicating trials with higher precision. The *p*-values for comparison with the reference group (shown as "ref") from the meta-regression analysis are also reported on the top of the bubble plots. \**P*-value < 0.05. AD, Alzheimer's Disease.
(PDF)

**S12 Fig. Bubble plot showing the results of meta-regression of the occurrence of ARIA-E, by (a) Mean age, (b) MMSE score, (c) ApoE4 carrier percentage, and (d) Amyloid beta burden in PET.** The size of the bubbles shows the inverse of the variance of the log-transformed risk ratio in each trial, with larger bubbles indicating trials with higher precision. The *p*-values from the meta-regression analysis are also reported. \**P*-value < 0.05. ARIA-E, amyloid-related imaging abnormalities with edema; MMSE, Mini-Mental State Examination; PET, positron emission tomography.
(PDF)

**S13 Fig. Bubble plots showing the results of meta-regression of the occurrence of ARIA-E, by (a) AD stage, (b) Drug, and (c) Antibody type.** The size of the bubbles shows the inverse of the variance of the log-transformed risk ratio in each trial, with larger bubbles indicating trials with higher precision. The *p*-values for comparison with the reference group (shown as "ref") from the meta-regression analysis are also reported on the top of the bubble plots. ARIA-E, amyloid-related imaging abnormalities with edema; AD, Alzheimer's Disease.
(PDF)

**S14 Fig. Bubble plots showing the results of meta-regression of the occurrence of ARIA-E, by (a) Binding mechanism and (b) Biological effect.** The size of the bubbles shows the inverse of the variance of the log-transformed risk ratio in each trial, with larger bubbles indicating trials with higher precision. The bubble plots on the left includes all trials, while the bubble plots on the right shows only trials with a risk ratio of up to 50. The *p*-values for comparison with the reference group (shown as "ref") from the meta-regression analysis are also reported on the top of the bubble plots. \**P*-value < 0.05. ARIA-E, amyloid-related imaging abnormalities with edema; AD, Alzheimer's Disease.
(PDF)

**S15 Fig. Bubble plots showing the results of meta-regression of the occurrence of headache, by (a) Mean age, (b) MMSE score, (c) ApoE4 carrier percentage, and (d) Amyloid beta burden in PET.** The size of the bubbles shows the inverse of the variance of the log-transformed risk ratio in each trial, with larger bubbles indicating trials with higher precision. The $p$-values from the meta-regression analysis are also reported. MMSE, Mini-Mental State Examination; PET, positron emission tomography.
(PDF)

**S16 Fig. Bubble plots showing the results of meta-regression of the occurrence of headache, by (a) AD stage, (b) Drug, (c) Antibody type, (d) Binding mechanism, and (e) Biological effect.** The size of the bubbles shows the inverse of the variance of the log-transformed risk ratio in each trial, with larger bubbles indicating trials with higher precision. The $p$-values for comparison with the reference group (shown as "ref") from the meta-regression analysis are also reported on the top of the bubble plots. AD, Alzheimer's Disease.
(PDF)

**S17 Fig. Bubble plots showing the results of meta-regression of the occurrence of fall, by (a) Mean age, (b) MMSE score, (c) ApoE4 carrier percentage, and (d) Amyloid beta burden in PET.** The size of the bubbles shows the inverse of the variance of the log-transformed risk ratio in each trial, with larger bubbles indicating trials with higher precision. The $p$-values from the meta-regression analysis are also reported. MMSE, Mini-Mental State Examination; PET, positron emission tomography.
(PDF)

**S18 Fig. Bubble plots showing the results of meta-regression of the occurrence of fall, by (a) AD stage, (b) Drug, (c) Antibody type, (d) Binding mechanism, and (e) Biological effect.** The size of the bubbles shows the inverse of the variance of the log-transformed risk ratio in each trial, with larger bubbles indicating trials with higher precision. The $p$-values for comparison with the reference group (shown as "ref") from the meta-regression analysis are also reported on the top of the bubble plots. AD, Alzheimer's Disease.
(PDF)

**S19 Fig. Bubble plots showing the results of meta-regression of the occurrence of dizziness, by (a) Mean age, (b) MMSE score, (c) ApoE4 carrier percentage, and (d) Amyloid beta burden in PET.** The size of the bubbles shows the inverse of the variance of the log-transformed risk ratio in each trial, with larger bubbles indicating trials with higher precision. The $p$-values from the meta-regression analysis are also reported. MMSE, Mini-Mental State Examination; PET, positron emission tomography.
(PDF)

**S20 Fig. Bubble plots showing the results of meta-regression of the occurrence of dizziness, by (a) AD stage, (b) Drug, (c) Antibody type, (d) Binding mechanism, and (e) Biological effect.** The size of the bubbles shows the inverse of the variance of the log-transformed risk ratio in each trial, with larger bubbles indicating trials with higher precision. The $p$-values for comparison with the reference group (shown as "ref") from the meta-regression analysis are also reported on the top of the bubble plots. AD, Alzheimer's Disease.
(PDF)

**S21 Fig. Forest plot for cerebral macrohemorrhage.**
(PDF)

**S22 Fig. Sensitivity analysis 1 (excluding solanezumab). Forest plot for efficacy outcomes.** (**a**) The Clinical Dementia Rating-Sum of Boxes (CDR-SB) and (**b**) Alzheimer's Disease Assessment Scale-Cognitive Subscale (ADAS-Cog).
(PDF)

**S23 Fig. Sensitivity analysis 1 (excluding solanezumab). Forest plots for safety outcomes.** (**a**) Death, (**b**) Serious Adverse Event, (**c**) ARIA-E, (**d**) ARIA-H. ARIA-E: Amyloid-Related Imaging Abnormalities-Effusion, ARIA-H, Amyloid-Related Imaging Abnormalities-Hemorrhage.
(PDF)

**S24 Fig. Sensitivity analysis 1 (excluding solanezumab). Forest plots for safety outcomes related to clinical symptoms.** (**a**) Headache, (**b**) Fall, and (**c**) Dizziness.
(PDF)

**S25 Fig. Sensitivity analysis 2 (excluding bapineuzumab). Forest plot for efficacy outcomes.** (**a**) The Clinical Dementia Rating-Sum of Boxes (CDR-SB) and (**b**) Alzheimer's Disease Assessment Scale-Cognitive Subscale (ADAS-Cog).
(PDF)

**S26 Fig. Sensitivity analysis 2 (excluding bapineuzumab). Forest plots for safety outcomes.** (**a**) Death, (**b**) Serious Adverse Event, (**c**) ARIA-E, (**d**) ARIA-H. ARIA-E: Amyloid-Related Imaging Abnormalities-Effusion, ARIA-H, Amyloid-Related Imaging Abnormalities-Hemorrhage.
(PDF)

**S27 Fig. Sensitivity analysis 2 (excluding bapineuzumab). Forest plots for safety outcomes related to clinical symptoms.** (**a**) Headache, (**b**) Fall, and (**c**) Dizziness.
(PDF)

**S28 Fig. Sensitivity analysis 3 (excluding aducanumab). Forest plot for efficacy outcomes.** (**a**) The Clinical Dementia Rating-Sum of Boxes (CDR-SB) and (**b**) Alzheimer's Disease Assessment Scale-Cognitive Subscale (ADAS-Cog).
(PDF)

**S29 Fig. Sensitivity analysis 3 (excluding aducanumab). Forest plots for safety outcomes.** (**a**) Death, (**b**) Serious Adverse Event, (**c**) ARIA-E, (**d**) ARIA-H. ARIA-E: Amyloid-Related Imaging Abnormalities-Effusion, ARIA-H: Amyloid-Related Imaging Abnormalities-Hemorrhage.
(PDF)

**S30 Fig. Sensitivity analysis 3 (excluding aducanumab). Forest plots for safety outcomes related to clinical symptoms.** (**a**) Headache, (**b**) Fall, and (**c**) Dizziness.
(PDF)

**S31 Fig. Sensitivity analysis 4 (excluding solanezumab and bapineuzumab). Forest plot for efficacy outcomes.** (**a**) The Clinical Dementia Rating-Sum of Boxes (CDR-SB) and (**b**) Alzheimer's Disease Assessment Scale-Cognitive Subscale (ADAS-Cog).
(PDF)

**S32 Fig. Sensitivity analysis 4 (excluding solanezumab and bapineuzumab). Forest plots for safety outcomes.** (**a**) Death, (**b**) Serious Adverse Event, (**c**) ARIA-E, (**d**) ARIA-H. ARIA-E, Amyloid-Related Imaging Abnormalities-Effusion; ARIA-H, Amyloid-Related Imaging Abnormalities-Hemorrhage.
(PDF)

**S33 Fig. Sensitivity analysis 4 (excluding solanezumab and bapineuzumab). Forest plots for safety outcomes related to clinical symptoms.** (**a**) Headache, (**b**) Fall, and (**c**) Dizziness.
(PDF)

**S34 Fig. Sensitivity analysis 5 (excluding solanezumab, bapineuzumab, and aducanumab). Forest plot for efficacy outcomes.** (**a**) The Clinical Dementia Rating-Sum of Boxes (CDR-SB) and (**b**) Alzheimer's Disease Assessment Scale-Cognitive Subscale (ADAS-Cog).
(PDF)

**S35 Fig. Sensitivity analysis 5 (excluding solanezumab, bapineuzumab, and aducanumab). Forest plots for safety outcomes.** (**a**) Death, (**b**)Serious Adverse Event, (**c**) ARIA-E, (**d**) ARIA-H. ARIA-E, Amyloid-Related Imaging Abnormalities-Effusion, ARIA-H, Amyloid-Related Imaging Abnormalities-Hemorrhage.
(PDF)

**S36 Fig. Sensitivity analysis 5 (excluding solanezumab, bapineuzumab, and aducanumab). Forest plots for safety outcomes related to clinical symptoms.** (a) Headache, (b) Fall, and (c) Dizziness.
(PDF)

**S37 Fig. Sensitivity analysis 6 (Low-dose populations). Forest plots for efficacy outcomes.** (**a**) The Clinical Dementia Rating-Sum of Boxes (CDR-SB) and (**b**) Alzheimer's Disease Assessment Scale-Cognitive Subscale (ADAS-Cog).
(PDF)

**S38 Fig. Sensitivity analysis 6 (Low-dose populations). Forest plots for safety outcomes.** (**a**) Death, (**b**) Serious Adverse Event, (**c**) ARIA-E, (**d**) ARIA-H. ARIA-E, Amyloid-Related Imaging Abnormalities-Effusion; ARIA-H, Amyloid-Related Imaging Abnormalities-Hemorrhage.
(PDF)

**S39 Fig. Sensitivity analysis 6 (Low-dose populations). Forest plots for safety outcomes related to symptoms.** (**a**) Headache, (**b**) Fall, and (**c**) Dizziness.
(PDF)

**S40 Fig. Sensitivity analysis 6 and 7 (Low and high dose populations). Forest plot for cerebral macrohemorrhage.**
(PDF)

**S41 Fig. Sensitivity analysis 7 (High-dose populations). Forest plots for efficacy outcomes.** (**a**) The Clinical Dementia Rating-Sum of Boxes (CDR-SB) and (**b**) Alzheimer's Disease Assessment Scale-Cognitive Subscale (ADAS-Cog).
(PDF)

**S42 Fig. Sensitivity analysis 7 (High-dose populations). Forest plots for safety outcomes.** (**a**) Death, (**b**) Serious Adverse Event, (**c**) ARIA-E, (**d**) ARIA-H. ARIA-E, Amyloid-Related Imaging Abnormalities-Effusion, ARIA-H, Amyloid-Related Imaging Abnormalities-Hemorrhage.
(PDF)

**S43 Fig. Sensitivity analysis 7 (High-dose populations). Forest plots for safety outcomes related to symptoms.** (**a**) Headache, (**b**) Fall, and (**c**) Dizziness.
(PDF)

**S44 Fig. Funnel plots for efficacy endpoints in sensitivity analysis 8 (including halted trials with sample size with fewer than 200 patients in each arm).** (**a**) The Clinical Dementia

Rating-Sum of Boxes (CDR-SB) and (**b**) Alzheimer's Disease Assessment Scale-Cognitive Subscale (ADAS-Cog). Filled circles represent estimated treatment effect (risk ratio) and its precision (standard error) for each individual study. In addition to individual study results, the fixed effect estimate (vertical dashed line) with 95% confidence interval limits (diagonal dashed lines) and the random effects estimate (vertical dotted line) are shown in the figures. Also, *p*-values of Egger's test are shown. \**P*-value < 0.05.
(PDF)

**S45 Fig.  Funnel plots for safety endpoints in sensitivity analysis 8 (including halted trials with a sample size of fewer than 200 patients in each arm).** (**a**) Death, (**b**) Serious adverse event, (**c**) ARIA-E, (**d**) ARIA-H. Filled circles represent estimated treatment effect (risk ratio) and its precision (standard error) for each individual study. In addition to individual study results, the fixed effect estimate (vertical dashed line) with 95% confidence interval limits (diagonal dashed lines) and the random effects estimate (vertical dotted line) are shown in the figures. Also, *p*-values of Egger's test are shown. \**P*-value < 0.05. ARIA-E, Amyloid-Related Imaging Abnormalities-Effusion, ARIA-H, Amyloid-Related Imaging Abnormalities-Hemorrhage.
(PDF)

**S46 Fig.  Funnel plots for safety endpoints in sensitivity analysis 8 (including halted trials with a sample size of fewer than 200 patients in each arm).** (**a**) Headache, (**b**) Fall, (**c**) Dizziness. Filled circles represent estimated treatment effect (risk ratio) and its precision (standard error) for each individual study. In addition to individual study results, the fixed effect estimate (vertical dashed line) with 95% confidence interval limits (diagonal dashed lines) and the random effects estimate (vertical dotted line) are shown in the figures. Also, *p*-values of Egger's test are shown. \**P*-value < 0.05. ARIA-E, Amyloid-Related Imaging Abnormalities-Effusion
(PDF)

**S47 Fig.  Sensitivity analysis 8 (including halted trials with a sample size of fewer than 200 patients in each arm).** Forest plots for the change in Clinical Dementia Rating-Sum of Boxes (CDR-SB).
(PDF)

**S48 Fig.  Sensitivity analysis 8 (including halted trials with a sample size of fewer than 200 patients in each arm).** Forest plots for the change in Alzheimer's Disease Assessment Scale-Cognitive Subscale (ADAS-Cog).
(PDF)

**S49 Fig.  Sensitivity analysis 8 (including halted trials with a sample size of fewer than 200 patients in each arm). Forest plots for safety outcomes.** (**a**) Death, (**b**) Serious adverse event.
(PDF)

**S50 Fig.  Sensitivity analysis 8 (including halted trials with a sample size of fewer than 200 patients in each arm). Forest plots for safety outcomes related to imaging abnormalities.** (**a**)ARIA-E, (**b**) ARIA-H. ARIA-E, Amyloid-Related Imaging Abnormalities-Effusion; ARIA-H, Amyloid-Related Imaging Abnormalities-Hemorrhage.
(PDF)

**S51 Fig.  Sensitivity analysis 8 (including halted trials with a sample size of fewer than 200 patients in each arm). Forest plots for safety outcomes related to symptoms.** (**a**) Headache, (**b**) Fall, (**c**) Dizziness.
(PDF)

## Author contributions

**Conceptualization:** Reina Tonegawa-Kuji, Feixiong Cheng.

**Data curation:** Reina Tonegawa-Kuji, Yuan Hou.

**Formal analysis:** Reina Tonegawa-Kuji, Yuan Hou, Bo Hu.

**Funding acquisition:** Feixiong Cheng.

**Investigation:** Reina Tonegawa-Kuji, Bo Hu, Babak Tousi, Feixiong Cheng.

**Methodology:** Reina Tonegawa-Kuji, Noah Lorincz-Comi, Feixiong Cheng.

**Project administration:** Feixiong Cheng.

**Resources:** Reina Tonegawa-Kuji, Andrew A. Pieper, James B. Leverenz, Feixiong Cheng.

**Software:** Reina Tonegawa-Kuji.

**Supervision:** Feixiong Cheng.

**Validation:** Reina Tonegawa-Kuji, Yuan Hou, Bo Hu, Andrew A. Pieper, Babak Tousi, James B. Leverenz, Feixiong Cheng.

**Visualization:** Reina Tonegawa-Kuji.

**Writing – original draft:** Reina Tonegawa-Kuji.

**Writing – review & editing:** Reina Tonegawa-Kuji, Andrew A. Pieper, Feixiong Cheng.

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
