## [Editor Report · Decision Letter 0]

18 Jul 2024

Dear Dr Cheng, 

Thank you for submitting your manuscript entitled "Systematic meta-analysis for passive immunotherapies in Alzheimer’s disease" for consideration by PLOS Medicine.

Your manuscript has now been evaluated by the PLOS Medicine editorial staff and I am writing to let you know that we would like to send your submission out for external peer review.

Please re-submit your manuscript within two working days, i.e. by Jul 22 2024.

Feel free to email me at atosun@plos.org or us at plosmedicine@plos.org if you have any queries relating to your submission.

Kind regards,

Alexandra Tosun, PhD

Associate Editor

PLOS Medicine

---

## [Decision Letter · Decision Letter 1]

20 Aug 2024

Dear Dr Cheng,

Many thanks for submitting your manuscript "Systematic meta-analysis for passive immunotherapies in Alzheimer’s disease" (PMEDICINE-D-24-02295R1) to PLOS Medicine. The paper has been reviewed by subject experts and a statistician; their comments are included below and can also be accessed here: [LINK]

As you will see, the reviewers were positive about the paper, but raised a number of questions for clarification, and made several suggestions for adjusting the analysis. After discussing the paper with the editorial team and an academic editor with relevant expertise, I'm pleased to invite you to revise the paper in response to the reviewers' comments. Please note that we may pass on comments from the academic editor by e-mail after the decision letter has been sent. We plan to send the revised paper to some or all of the original reviewers, and we cannot provide any guarantees at this stage regarding publication.

We ask that you submit your revision by Sep 10 2024. However, if this deadline is not feasible, please contact me by email, and we can discuss a suitable alternative.

Don't hesitate to contact me directly with any questions (atosun@plos.org). 

Best regards, 

Alexandra 

Alexandra Tosun, PhD 

Associate Editor

PLOS Medicine

atosun@plos.org

Comments from the reviewers: 

Reviewer #1: The authors perform meta-analysis of passive immunotherapies in Alzheimer's disease. Their results show that the passive immunotherapies slow cognitive decline with modest effect sizes. However, the therapies are also questioned for serious adverse effects of ARIA and headaches. The study and finding are interesting, which can add to therapeutic strategy development for AD. However, there are some concerns need to be addressed by the authors:

1, there are multiple clinical trial studies on distinct anti-AB antibodies, however, the authors finally make a conclusion on effects of passive immunotherapies for AD. As showed in table 3, some antibodies fail to have protective effects. So based on some successful and fail studies, how the final conclusions can be arrived?

2, in table 3, how the authors get the p value? whether they have analyzed based on the original data from respective studies? 

3, There are two primary versions of the ADAS-Cog: ADAS-Cog 11 (Original version) and ADAS-Cog 13 (Extended version), so how the authors have done to solve the difference of ADAS-Cog 11 and 13 in different studies?

Reviewer #2: a) The methodology section is comprehensive, but it may benefit from more explicit detailing of the meta-regression models used. Specifically, describing how each potential moderator (e.g., age, MMSE, APOE4 status) was selected for inclusion in the model could add clarity.

b) The p-value for ADAS-Cog (Age) reported in the text does not match the value provided in Figure S4 (0.82 vs. 0.84). Please verify and correct this discrepancy.

c) Please provide meta-regression graphs for all parameters listed in Table 2, similar to those presented in Figures S3 and S4. This is particularly important for parameters with statistically significant results, such as AD stage and Binding mechanism.

d) In Table 2, it is mentioned that the use of Bapineuzumab, compared to Lecanemab, was associated with a significant negative impact on the CDR-SB (p=0.03), with no significant association between drugs and ADAS-Cog. Please clarify this point in the table, and, if possible, provide corresponding graphs as mentioned earlier.

e) Please include the results (mean difference, confidence interval, and p-value) for all parameters listed in Table 2 in Table 3 for consistency.

f) I suggest consolidating the base-case and sensitivity analysis results into a single table. Including confidence intervals for mean difference, standardised mean difference, relative risk, and p-values would enhance comparability.

g) The sensitivity analyses performed are robust, but an additional analysis that excludes trials with high heterogeneity might further strengthen the conclusion

h) The manuscript contains several forest plots and other figures that effectively communicate the results. Consider suggesting that the authors provide funnel plots for the secondary outcomes as well, similar to those provided for the primary outcomes

i) The limitations section is well-addressed, but it might be useful to discuss more about the potential impact of excluding smaller trials (those with <200 participants) on the overall findings. The authors mention a possible exclusion of smaller trials with favourable outcomes for mAb therapies; expanding on how this might bias the results would be insightful.

j) Given the mention of medico-economic aspects in the discussion, it might be beneficial to recommend that the authors include a brief section or a supplemental table summarising the cost implications of the treatments evaluated, even if only qualitatively. This would add value for readers interested in the broader impact of these therapies.

k) The data availability statement is clear, but consider suggesting that the authors provide more detailed information on how the data were processed and analysed, possibly through an online repository or supplementary materials.

Reviewer #3: This manuscript summarized and analyzed outcomes and side effects of antibodies against Aβ from 13 phase III clinical trials. I have several suggestions to improve the quality of this study.

1. Because the authors only analyzed results from Aβ antibody drugs, the following title may be more suitable for this paper : "Systematic meta-analysis for passive immunotherapies targeting amyloid beta in Alzheimer's disease".

2. The effect of anti- Aβ immunotherapies on fluid biomarkers and tau pathology should be analyzed.

3. Please discuss why anti- Aβ immunotherapies only have modest effect on slowing cognitive decline.

Reviewer #4: Tonegawa-Kuji et al provide a meta-analytical approach of the recently complete Phase 3 studies of amyloid immunotherapy to include recently published studies from donanemab. By including the CDR-SB and the ADAS-Cog across all studies, they have evaluated the overall clinical and safety effects of these therapies, as well as assessing the effects of soluble/monomer targeting or insoluble/aggregated targeting therapies. Overall, the findings are consistent with the individual trials suggesting modest clinical benefits and a clear association with brain edema/hemorrhages. The inclusion of the meta-regression analyses is a nice addition to previously similar meta-analyses, however the grouping of all therapies together for these analyses likely results in a some inaccuracies related to some of the variables association with the outcomes; this was partially addressed with the sensitivity analyses provided.

I have only a few comments for consideration:

Methods:

How did the authors derive the level of effect based on the SMD for ADAS-Cog and why was a similar effect size not proposed for CDR-SB? There are studies published on the typical annual rates of change of the CDR-SB that could be used for consideration of meaningful effect sizes.

For the Aducanumab studies, how did the authors account for the change in dosing for APOE 4 carriers during the trial (difference in the maximal dose allowed)? 

Results:

The serious AE outcomes seem to counter common sense? Given the clear absolute risk of ARIA events with immunotherapy and the association of ARIA with SAEs, it seems that these events should indicate a higher risk. This is probably a result of aggregating all the trials together, but it even in those trials that have reported significant SAEs from amyloid reduction, these are not clear here.

This sentence on page 10 seems to be opposite of the statistics: 'There was a statistically significant interaction between the binding mechanism of the antibody and the risk of ARIA-E (p=0.005), indicating a lower risk of observing AIRA-E in trials that used antibodies that preferentially bind to Ab aggregates or plaque compared to those that bind Ab monomers' . The way this is written, it seems that ab aggregate binding have less ARIA than monomers, which is clearly not the case.

It would be helpful to include a number needed treat type analyses including a number needed to harm to better translate these findings to patient level data.

Reviewer #5: This manuscript, as stated by the authors, is an extension of previous publication, and can potentially provide additional supportive evidence regarding the effectiveness of anti-amyloid treatment. I would greatly appreciate if the author can address the following comments.

1. One major issue is the conclusion in the discussion section. It states that" 5) compared to Lecanemab, other antibodies were as effective in slowing the cognitive decline, except for Bapineuzumab". This statement has to be supported by non-inferiority test or equivalence test, however, all the tests in this manuscript look like two-sided, testing-for-equality, tests.

2. Please provide details about how SMD was calculated.

3. Please provide details about how common-effect model and random-effect model, can be put into supplementals. When random-effect model is abbreviated as REML? 

4. In these analyses:" the following variables impacted the results: age, baseline cognitive scores on mini-mental state examination (MMSE), percentage of APOE4 carriers included in trials…", why the baseline CDR-SB is not used as a covariate when analyzing the change in CDR-SB, similarly, why ADAS-Cog is not used as a covariate when analyzing ADAS-cog? This is against all the typical analysis for AD trials.

5. "Follow-up periods ranged from 76 to 116 months, with the majority being 78 months.", they are not months, but weeks? For Clarity AD, they reported 18 months, it's not clear it's 78 weeks or 72 weeks, so just use 18 months for Clarity AD. 

6. Table 3 compared early-stage AD vs mild-moderate, however, it is not clear which trials are included in the early-stage AD group and which ones in the mild-moderate group, please clarify. 

7. It states that:" In the main analysis, the groups with two doses of the antibody arm were combined to form a single antibody group,", it's not clear how they were combined to get the average treatment effect in CDR-SB/ADAS-cog? Please provide details.

8. Is this:" Aducanumab and Lecanemab were approved based on their effect on Ab reduction and not their effects on improving cognition" accurate? 

9. What does this mean:" Other than Bapineuzumab, there was no association between drugs and efficacy endpoints."?

10. "However, in the open-label Lecanemab study, three deaths from brain hemorrhage were reported, suggesting that close patient monitoring is warranted (23)", how about the death occurred in the donanemab phase 3 trial, should not those be included and discussed?

---

* Please upload any figures associated with your paper as individual TIF or EPS files with 300dpi resolution at resubmission; please read our figure guidelines for more information on our requirements: http://journals.plos.org/plosmedicine/s/figures. While revising your submission, please upload your figure files to the PACE digital diagnostic tool, https://pacev2.apexcovantage.com/. PACE helps ensure that figures meet PLOS requirements. To use PACE, you must first register as a user. Then, login and navigate to the UPLOAD tab, where you will find detailed instructions on how to use the tool. If you encounter any issues or have any questions when using PACE, please email us at PLOSMedicine@plos.org.

* FINANCIAL DISCLOSURES: Please state whether any sponsors or funders (other than the named authors) played any role in study design, data collection and analysis, the decision to publish, or preparation of the manuscript. If they had no role in the research, include this sentence: “The funders had no role in study design, data collection and analysis, decision to publish, or preparation of the manuscript.”

FIGURES AND TABLES

SUPPLEMENTARY MATERIAL

REFERENCES

* Where website addresses are cited, please include the complete URL and specify the date of access (e.g. [accessed: 12/06/2024]).

STUDY TYPE-SPECIFIC REQUESTS 

* Please report your SR/MA according to the PRISMA guidelines provided at the EQUATOR site. http://www.equator-network.org/reporting-guidelines/prisma/. Please provide the completed PRISMA checklist as Supporting Information. When completing the checklist, please use section and paragraph numbers, rather than page numbers. Please add the following statement, or similar, to the Methods: "This study is reported as per the Preferred Reporting Items for Systematic Reviews and Meta-Analyses (PRISMA) guideline (S1 Checklist)." 

* Abstract: Please report your abstract according to PRISMA for abstracts (https://doi.org/10.1371/journal.pmed.1001419) following the PLOS Medicine abstract structure (Background, Methods and Findings, Conclusions). Please ensure you provide dates of search, data sources, number of studies included, types of study designs included, eligibility criteria, and synthesis/appraisal methods.

* Please note that we expect searches to be updated to within 6 months of the time of submission.

---

## [Decision Letter · Decision Letter 2]

26 Nov 2024

Dear Dr Cheng,

Many thanks for re-submitting your manuscript "Systematic review and meta-analysis for passive immunotherapies targeting amyloid beta in Alzheimer’s disease" (PMEDICINE-D-24-02295R2) for review by PLOS Medicine. The paper has been seen again by three subject experts and a statistician; their comments are included below and can also be accessed here: [LINK]

Thank you for your detailed response to the reviewers' comments. As you will see, the reviewers are mostly satisfied with your responses to their comments. There are a number of remaining comments from the statistical reviewer that require further clarification. After discussing the paper with the editorial team, we ask you to carefully address the comments in a further revision. We plan to send the revised paper to some or all of the original reviewers.

We ask that you submit your revision by Dec 17 2024. However, if this deadline is not feasible, please contact me by email, and we can discuss a suitable alternative.

Don't hesitate to contact me directly with any questions (atosun@plos.org). 

Best regards, 

Alexandra 

Alexandra Tosun, PhD 

Associate Editor

PLOS Medicine

atosun@plos.org

Comments from the editorial team:

Please note that we require a point-by-point response to not only reviewer comments, but all editorial comments, including general editorial requests. Please be sure to provide such a document that addresses the editorial comments of the previous decision letter.

Comments from the reviewers: 

Reviewer #2: 1- It would be helpful to provide a clearer statement on the assumptions underlying the meta-regression, particularly regarding potential inter-study variability in moderator definitions. 

2- While the "meta" package was chosen for the analysis, the manuscript does not explain why this specific package was selected over other options, such as "metafor," which may offer alternative estimators or variance structures for meta-regression. Please provide a rationale for selecting the "meta" package, possibly including a brief comparison to other methods or tools, to ensure that this choice yields the most reliable results for the data used.

3- Egger's test and funnel plots are utilized to assess publication bias for certain outcomes, but it remains unclear how selective reporting was addressed, particularly regarding smaller trials with fewer than 200 participants, which were excluded. It would be valuable to discuss any potential impact of excluding these smaller studies on publication bias and to describe any steps taken to mitigate this bias. 

4- Trials like those for Aducanumab were excluded based on interim or incomplete data. However, excluding these trials entirely may overlook potentially valuable emerging evidence. Please discuss any impact this exclusion may have on the conclusions. Alternatively, consider including a separate analysis that cautiously incorporates these interim data to examine their influence on the pooled estimates.

5- The manuscript does not provide details on how missing data were managed within the meta-regression analysis. Given that some trials reported different outcomes or utilized distinct scales (e.g., ADAS-Cog versions 11, 13, and 14), this could introduce inconsistencies in the results if not addressed adequately. Please describe any imputation methods or sensitivity analyses conducted for missing data. If missing data were not specifically addressed, consider including this as a limitation, as it could impact the meta-regression's robustness.

6- There are some inconsistencies in format and reporting of confidence intervals and p-values. Please review the manuscript for consistency in reporting statistical significance and confidence intervals (e.g., using either 95% CI consistently throughout).

7- Although heterogeneity is assessed and reported, the manuscript could benefit from a more detailed discussion of what specific levels of heterogeneity imply for the interpretation of results in this context. Consider adding a brief section discussing the interpretation of heterogeneity levels (e.g., low, moderate, high) and their potential influence on the reliability of pooled estimates. 

8- The NNT/NNH calculations are provided using Kraemer's method, but this approach is known to yield conservative estimates. Although the text mentions an alternative method (Furukawa's), the reasoning behind choosing Kraemer's approach is only briefly stated. Please provide a more in-depth explanation of why Kraemer's method was chosen over other methods for NNT/NNH calculations, despite its conservative estimates. Additionally, including a brief summary of how Furukawa's method might yield different results could contextualize the limitations of using Kraemer's method in this context.

9- In some tables, such as Supplemental Tables S2-S5, there are inconsistencies in data presentation (e.g., varying decimal places or differing levels of detail). Please standardize data presentation in all supplemental tables (e.g., using two decimal places throughout or providing consistent detail for each entry). 

10- Supplemental Text S3 outlines the methodology for combining groups in cases with multiple doses, yet it could be challenging for readers to follow without more detailed steps or examples. Please provide a step-by-step example of how multiple doses were combined to form a single group for analysis.

11- The summary of neuroimaging and biomarker results in Supplemental Tables S3 and S4 is helpful, but lacks specific details on study populations, methodologies, or outcome variability, which could affect the interpretation of results. If possible, add details on the neuroimaging methodologies (e.g., PET tracer types) and biomarker assays used across trials, as well as any notable variations in outcomes.

12- While funnel plots and Egger's tests are presented for primary outcomes, a similar assessment for secondary outcomes would strengthen confidence in these findings.

Reviewer #4: Thank you for the thorough response and updated data. I have no further comments at this time.

Reviewer #5: I appreciate the authors' efforts to address all the comments.

---

## [Decision Letter · Decision Letter 3]

5 Feb 2025

Dear Dr. Cheng,

Thank you very much for re-submitting your manuscript "Systematic review and meta-analysis for passive immunotherapies targeting amyloid beta in Alzheimer’s disease" (PMEDICINE-D-24-02295R3) for review by PLOS Medicine.

I have discussed the paper with my colleagues and the academic editor and it was also seen again by the statistical reviewer. I am pleased to say that provided the remaining editorial and production issues are dealt with we are planning to accept the paper for publication in the journal.

[LINK]

In revising the manuscript for further consideration here, please ensure you address the specific points made by the editors. In your rebuttal letter you should indicate your response to the editors' comments and the changes you have made in the manuscript. Please submit a clean version of the paper as the main article file. A version with changes marked must also be uploaded as a marked up manuscript file.

We ask that you submit your revised manuscript within 1 week (February 12). Please email me directly (atosun@plos.org) if you have any questions or concerns or if you need to request an extension to the resubmission deadline.

We look forward to receiving the revised manuscript. 

Kind regards,

Heather

Heather Van Epps, PhD

Executive Editor

[on behalf of]

Alexandra Tosun, PhD

Senior Editor 

PLOS Medicine

plosmedicine.org

Comments from the academic editor:

1. I would suggest they stick to “small” rather than “modest” effects. 

2. The only area of concern for me is the suggestion that the drug price is somehow cost effective. The NICE analysis suggests even if the small effect size is considered to be meaningful (which is contested), the cost would be miles off what is considered good value, including the challenge of the side effects. The authors mention no effect on cerebral haemorrhage but the ARIA side effects are haemorrhagic so that is a bit misleading.

Comments from the editors:

1. Title: please modify the title so it complies with PLOS Medicine's style. Your title must be nondeclarative and not a question. It should begin with main concept if possible. "Effect of" should be used only if causality can be inferred, i.e., for an RCT. Please place the study design ("A randomized controlled trial," "A retrospective study," "A modelling study," etc.) in the subtitle (ie, after a colon). We suggest modifying your title to: “Efficacy and safety of passive immunotherapies targeting amyloid beta in Alzheimer’s disease: a systematic review and metaanalysis” (or similar).

2. Please confirm that your abstract complies with our requirements, including providing all the information relevant to this study type https://journals.plos.org/plosmedicine/s/submission-guidelines#loc-abstract

3. Abstract, line 38: please modify “sporadic AD patients” to “patients with sporadic AD” and ensure that you use similar patient-centric language throughout the manuscript (eg, abstract line 47-48 and other instances throughout the paper). 

4. Abstract, lines 39-40: please provide the beginning and end dates of your search (eg, database inception to Jan 4, 2024?). This should also be added to the methods section.

5. Abstract, line 47: please include the comparison within the sentence; eg, “Thirteen trials (>18,000 patients) revealed an overall improvement with mAb treatment compared to placebo, with a mean difference in CDR-SB of -0.25 (95%CI [-0.38, -0.11]) (improvement) and a standardized mean difference in ADAS-Cog of -0.09 (95%CI [-0.12, -0.06]).

6. Abstract: please include a sentence at the end of the Methods and Findings section indicating the main limitation(s) of the study. 

7. Abstract: please specify that there were no language restrictions applied to your search. 

8. Drug names (donanemab, lecanemab, etc) should not be capitalized; please modify throughout the paper. 

9. Please remove numerical headings and sub-headings (ie, use only text in headings).

10. Figures. Please consider avoiding the use of red and green in order to make your figures more accessible. 

11. Please resupply the PRISMA checklist using section and paragraph numbers, rather than line numbers.

Comments from Reviewers:

Reviewer #2 (statistics): 

Thank you for thoroughly addressing all my comments and providing detailed clarifications in your revised manuscript. I appreciate the effort you have put into enhancing the rigor and transparency of your analyses.

The manuscript now reflects a robust and valuable contribution to subject, I am happy to recommend this work for publication and look forward to seeing its positive impact in the field.

[LINK]

---

## [Editor Report · Decision Letter 4]

21 Feb 2025

Dear Dr. Cheng,

Thank you very much for re-submitting your manuscript "Efficacy and safety of passive immunotherapies targeting amyloid beta in Alzheimer's disease: a systematic review and meta-analysis" (PMEDICINE-D-24-02295R4) for review by PLOS Medicine.

There are a few minor editorial issues that need to be addressed before we can accept the manuscript for publication; these are outlined at the end of this email. Please revise the paper accordingly, and submit the final revision within 1 week (Feb 28 2025).

Please ensure you address the specific points made by the editors. In your rebuttal letter you should indicate your response to the editors' comments and the changes you have made in the manuscript. Please submit a clean version of the paper as the main article file. A version with changes marked must also be uploaded as a marked up manuscript file. Please also check the guidelines for revised papers at http://journals.plos.org/plosmedicine/s/revising-your-manuscript for any that apply to your paper.

A reminder that when your manuscript is accepted, an uncorrected proof of your manuscript will be published online ahead of the final version, unless you've already opted out via the online submission form. If, for any reason, you do not want an earlier version of your manuscript published online or are unsure if you have already indicated as such, please let the journal staff know immediately at plosmedicine@plos.org.

If you have any questions in the meantime, please contact me directly at atosun@plos.org.

We look forward to receiving the revised manuscript.

Sincerely,

Alexandra Tosun, PhD

Associate Editor

PLOS Medicine

Requests from Editors:

1) Abstract: Please define ‘CI’ at first use.

2) Abstract: “Early-stage AD patients” – please revise with regard to use of patient-centered language (i.e., patients with early-stage AD).

3) Statistical reporting:

* Please report statistical information as follows to improve clarity for the reader "HR 2.00 (95% CI [1.99,2.01])".

* Please define statistical abbreviations at first use.

* Please repeat statistical definitions (HR, CI etc.) for each set of parentheses.

4) Author Summary: Please introduce the abbreviation ‘AD’ in the first bullet point or spell out Alzheimer’s disease throughout the Author Summary.

5) Author Summary: Please change ‘AD patients’ to ‘patients with AD’.

6) Author Summary: In the final bullet point of 'What Do These Findings Mean?', please include the main limitations of the study in non-technical language.

7) Results: On line 228, please change ‘AD patients’ to ‘patients with AD’.

8) Table 2: “only the mild AD patients were included” – please change to: “only patients with mild AD were included”.

9) Table 3/Table 4: Please define ‘Ab’, ‘PET’, ‘NA’, ‘CI’ below the table.

10) Discussion: On lines 467 and 468, please change ‘early AD patients’ to ‘patients with early AD’.

11) Figure 1: Please define ‘AD’ and RCT’ in the figure description.

12) Figure 5: Please define ‘RR’ and ‘CI’ in the figure description.

---

## [Editor Report · Decision Letter 5]

25 Feb 2025

Dear Dr Cheng, 

On behalf of my colleagues and the Academic Editor, Carol Brayne, I am pleased to inform you that we have agreed to publish your manuscript "Efficacy and safety of passive immunotherapies targeting amyloid beta in Alzheimer's disease: a systematic review and meta-analysis" (PMEDICINE-D-24-02295R5) in PLOS Medicine.

I appreciate your thorough responses to the reviewers' and editors' comments throughout the editorial process.

PRESS

Thank you again for submitting to PLOS Medicine. We look forward to publishing your manuscript.

Sincerely, 

Alexandra Tosun, PhD 

Associate Editor 

PLOS Medicine